# Mnemosyne: Learning to Train Transformers with Transformers

**Deepali Jain** *
Google DeepMind,
jaindeepali@google.com

**Krzysztof Choromanski** *
Google DeepMind and Columbia University,
kchoro@google.com

**Avinava Dubey** *
Google Research,
avinavadubey@google.com

**Sumeet Singh**
Google DeepMind,
ssumeet@google.com

**Vikas Sindhwani**
Google DeepMind,
sindhwani@google.com

**Tingnan Zhang**
Google DeepMind,
tingnan@google.com

**Jie Tan**
Google DeepMind,
jietan@google.com

## Abstract

In this work, we propose a new class of learnable optimizers, called *Mnemosyne*. It is based on the novel spatio-temporal low-rank implicit attention Transformers that can learn to train entire neural network architectures, including other Transformers, without any task-specific optimizer tuning. We show that Mnemosyne: (a) outperforms popular LSTM optimizers (also with new feature engineering to mitigate catastrophic forgetting of LSTMs), (b) can successfully train Transformers while using simple meta-training strategies that require minimal computational resources, (c) matches accuracy-wise SOTA hand-designed optimizers with carefully tuned hyper-parameters (often producing top performing models). Furthermore, Mnemosyne provides space complexity comparable to that of its hand-designed first-order counterparts, which allows it to scale to training larger sets of parameters. We conduct an extensive empirical evaluation of Mnemosyne on: (a) fine-tuning a wide range of Vision Transformers (ViTs) from medium-size architectures to massive ViT-Hs (36 layers, 16 heads), (b) pre-training BERT models and (c) soft prompt-tuning large 11B+ T5XXL models. We complement our results with a comprehensive theoretical analysis of the compact associative memory used by Mnemosyne which we believe was never done before.

## 1 Introduction

Learning-to-learn (L2L) systems [66, 63, 4, 5, 52, 28, 60, 74, 2, 12] used to train machine learning (ML) optimizers can be thought of as a natural lifting (to the optimizer-space) of the idea that has revolutionized ML: replacing hand-engineered features with the learned ones.

The idea to train ML optimizers is tempting, yet the lift to the optimizer-space comes at a price: the instances used to train such systems are the optimization problems themselves. Generalization in such a setting means: the ability to transfer knowledge to "similar" optimization tasks not seen in training. Rigorous mathematical analysis of the properties of L2L systems, that involves defining distributions over optimization problems, becomes challenging and is a subject on its own. Indeed, the literature on *meta-learning* is voluminous [73, 6, 53, 76, 1, 39, 24] and of critical importance in many disciplines such as Robotics, where transfer knowledge from simulator to hardware is a notoriously difficult problem [42, 32, 57].

---

*equal contribution

37th Conference on Neural Information Processing Systems (NeurIPS 2023).

A standard approach to learning optimizers is to cast it as a *sequential decision problem*, where a function $f$ called an *optimizee* is optimized via another function $g_\theta$ (an optimizer) with learnable parameters $\theta$. Function $f$ can take as input $\mathbf{x}$ the parameters of a neural network (NN) and output its corresponding test loss on a given task. The optimizer $g_\theta$ updates $\mathbf{x}$ as:

$$\begin{cases} \mathbf{x}_0 \leftarrow \text{optimization initial point}, \\ \mathbf{x}_{t+1} = g_\theta(f, \mathbf{x}_0, ..., \mathbf{x}_t) & \text{if } t > 0 \end{cases} \tag{1}$$

Function $g_\theta$ is either trained by minimizing the meta-loss objective $\mathcal{L}_\theta$, which is usually a sum of $f$-losses over certain time horizons, with supervised/reinforcement learning [40] or in a completely supervised way, where it learns to imitate expert-optimizers [13]. It usually does not act directly on the sequence $(\mathbf{x}_0, ..., \mathbf{x}_t)$, but its processed-version, e.g. the sequence of the corresponding gradients $(\nabla f(\mathbf{x}_0), ..., \nabla f(\mathbf{x}_t))$ if $f$ is differentiable (which does not need to be the case [50]). In practice, the processed-versions often have much richer structure [49]: *"...various rolling statistics including...loss data structures, the rolling momentum / rms terms, as well as the gradient clipping state..."*.

In this paper, we abstract from the low-level design of the sequential inputs to $g_\theta$ (which is a subject on its own) and different training strategies of $g_\theta$. Our interest is in the core design of $g_\theta$ since it acts as a memory-based system. Indeed, most models of $g_\theta$ leverage recurrent NN cells, such as LSTMs [2, 58, 57], that keep the history of the optimization-rollout in the form of a compact learnable latent state. In addition, due to the fact that inputs $\mathbf{x}$ to $f$ are usually high-dimensional (e.g. neural networks' parameter-vectors), $g_\theta$ is often factorized to process independently different dimensions of $\mathbf{x}$ [2].

The Transformer-revolution in ML set the stage for a very different way to model sequential data and in general: to model memory - the *attention mechanism* [67, 19, 54, 8, 22, 11]. It is natural to ask whether attention architectures can be used to replace LSTM memory cells in L2L systems. Applying Transformers here is compelling - modeling long-range relationships over time by avoiding catastrophic forgetting (which is what LSTMs struggle with, but Transformers are particularly good at) is especially relevant for optimizing highly non-convex objectives in deep learning. Yet these benefits come at the cost of quadratic (in the sequence-length) space and time complexity.

**Contributions.** We propose a new class of learnable optimizers, called *Mnemosyne*. It is based on the novel spatio-temporal low-rank implicit attention Transformers that can learn to train entire neural network architectures, including other Transformers, without any task-specific optimizer tuning. We show that Mnemosyne: (a) outperforms popular LSTM optimizers (also with new feature engineering to mitigate catastrophic forgetting of LSTMs), (b) can successfully train Transformers while using simple meta-training strategies leveraging minimal computational resources, (c) matches accuracy-wise SOTA hand-designed optimizers with carefully tuned hyper-parameters (often producing top performing models). As we show in Sec. 5, SOTA hand-designed optimizers are very sensitive to the hyperparameter choice: a hyperparameter selection that is optimal for one dataset might perform very poorly on another one. Thus the ability of the optimizer to automatically "implicitly learn" optimal hyperparameters is of critical practical importance. Furthermore, Mnemosyne provides space complexity comparable to that of its standard hand-designed first-order counterparts, which allows it to scale to training larger sets of parameters. We conduct an extensive empirical evaluation of Mnemosyne on: (a) fine-tuning a wide range of Vision Transformers (ViTs) from medium-size architectures to massive ViT-Hs (36 layers, 16 heads), (b) pre-training BERT models and (c) soft prompt-tuning large 11B+ T5XXL models. We also conduct a thorough theoretical analysis of the compact associative memory used by Mnemosyne, to the best of our knowledge never done before.

Mnemosyne leverages several algorithmic techniques: (a) efficient Transformers, called *Performers* [18], applying implicit low-rank attention and guaranteeing linear (in the history length for causal and parameter-tensor size for the spatial attention) time and space complexity, (b) bi-directional and uni-directional attention combined in the unified spatio-temporal system, (c) hierarchical spatial attention mechanism to further reduce memory footprint of the spatial attention encoders that can be thought of as a novel attention pooling mechanism (see: 3.2.1). Obtained L2L mechanism effectively acts as a *compact associative memory* (CAM) (see: Sec. 4) fed with latent representations summarizing groups of parameters induced by the natural structure of the target model to be trained (the so-called *topological encodings*, see: Sec 3.2). It thus provides the best of both worlds: efficiency due to the compact fixed-size hidden state (as in LSTMs) and expressiveness since this hidden state approximates regular Transformer's attention via the CAM-mechanism. We also believe that this paper lays the groundwork for the research on general-purpose attention-based learnable optimizers and foundational models for L2L systems.

## 2 Related work

The research on L2L systems involves a plethora of techniques: curriculum learning (e.g. incremently increasing optimizer's unroll length [13]), randomly scaled optimizees in training with relative scaling of input gradients [47], hierarchical RNN-architectures with lower memory and compute overhead that are capable of capturing inter-parameter dependencies [71] and more [9, 51, 15, 68].

Regular Transformers are recently considered in this setting, in particular to tune hyperparameters [16], to learn BFGS-type optimization for motion reconstruction [26] or as memory-systems in class-incremental learning [31]. Scalable Transformers [65, 64] were designed to address computational limitations of their regular counterparts. Several efficient attention mechanisms were proposed, based on hashing [35], clustering [59], dimensionality reduction [69] or sparsity [75].

In this paper, we apply in particular methods approximating attention via low-rank decomposition of the attention matrix [18, 44, 46, 17, 43, 20], due to their intrinsic connection with associative memory and energy-models [36] (while used in the causal attention setting). Regular Transformers can be thought of as differentiable dictionaries applying powerful *associative memory mechanisms* [37, 56], i.e. modern Hopfield networks [29] with exponential memory. Linear low-rank attention mechanisms are their compact variants [61] with intriguing theoretical properties and as such are perfect candidates for scalable memory-systems with respect to the temporal axis (Mnemosyne addresses also the problem of the efficient spatial encodings of the trainable parameters, see Sec. 3.2). That interpretation has profound practical consequences as giving guidance on the optimal version of the low-rank attention mechanism. Mnemosyne uses the so-called *hyperbolic cosine random features* (see: Sec. 3.3.1, A.1) providing particularly low variance of the softmax-kernel estimation, a key ingredient of those modern associative memory models.

## 3 Learning to learn with spatio-temporal attention

In this section, we give the description of Mnemosyne. Since the system consists of several components, we start by providing a high-level overview. We then discuss individual components in more depth: topological (spatial) encoder in Sec. 3.2 and temporal (causal) encoder in Sec. 3.3 .

### 3.1 Preliminaries: tree-like optimization domains

Consider an optimizee $f : \mathcal{D} \to \mathbb{R}$. In the simplest case, we can take: $\mathcal{D} \subseteq \mathbb{R}^d$ for $d \in \mathbb{N}_+$. However it will be convenient to impose a tree-like structure on the elements $\mathbf{x} \in \mathcal{D}$. We will assume that there exists a tree $\mathcal{T}_f$ such that for every $\mathbf{x} = (x_1, ..., x_d)^\top \in \mathcal{D}$, the set $\{x_1, ..., x_d\}$ is partitioned into non-empty subsets $S_1, ..., S_l$ corresponding to different leaves of $\mathcal{T}_f$, where $l$ stands for their number. Not only does $\mathcal{T}_f$ define partitioning of $\{x_1, ..., x_d\}$ through the subsets residing in different leaves, but it also imposes a natural hierarchy. The construction might seem artificial at first glance, but we have a good reason to introduce it early on - it is a predominant description of the NN structure (our main interest in this paper is to optimize NNs with learnable optimizers thus we identify elements of $\mathcal{D}$ with different instantiations of the particular NN model). In this context, different leaves correspond to individual tensors of parameters (e.g. weight-matrices or bias-vectors; here subsets $\mathcal{S}_i$ are just their flattened representations) and tree-induced hierarchy describes how the NN layers/modules emerge from those individual tensors [2]. This more general framework covers also the setting $\mathcal{D} \in \mathbb{R}^d$, where $\mathcal{T}_f$ is a star-tree with different leaves corresponding to different variables $x_i$.

### 3.2 Topological encoder

Standard L2L systems [2] act independently on the individual parameters to be optimized. We refer to this strategy as a ***coordinate-wise*** approach. It has the advantage of enabling lots of data for meta-training (since the optimization trajectory of each scalar parameter is a viable training point) and corresponds to $\mathcal{T}_f$ being a star-tree (see: Sec. 3.1). However it comes at a price of the memory footprint since space complexity becomes linear in the total number of trainable parameters. Mnemosyne can be successfully applied in the coordinate-wise framework (see: Sec. 5.2), but supports an arbitrary $\mathcal{T}_f$, in particular the natural variant, where leaves correspond to the individual tensors of the NN to be optimized and that we refer to as the ***tensor-wise*** approach.

Mnemosyne acts independently on each tensor $\mathbf{T}$ from each leaf of a given input $\mathcal{T}_f$. Tensor $\mathbf{T}$ is first flattened/vectorized and a representation vector $\mathbf{r}$ is associated with each parameter of $\mathbf{T}$. Since

---

[2] An example of the instantiation of this mechanism is a `pytree` structure that is a default representation of the neural network parameters used in JAX [25] - a popular machine learning framework used to transform numerical functions and train neural networks.

in this paper we minimize feature engineering for the L2L systems, our default choice for $\mathbf{r}$ is the 2-dimensional vector of the absolute value of the gradient dimension and its sign (that was used on a regular basis in several papers on the subject), but we emphasize that Mnemosyne is agnostic to the particular representation choice. The resulting sequence of representations $\mathcal{S}_{\mathbf{T}}$ is then transformed by the bi-directional attention of the Performer model [18]. A latent encoding of a fixed token from that sequence is then output as a *topological encoding* of $\mathbf{T}$.

### 3.2.1 Compactifying topological encoder with the hierarchical pooling

Using bi-directional linear attention from Performers is justified by the fact that sequence $\mathcal{S}_{\mathbf{T}}$ can be in practice very long. In fact it can easily surpass $1M$ tokens (for instance if $\mathbf{T}$ is a square weight-tensor corresponding to two consecutive layers of size $> 1K$ each). In that setting, even linear attention might not suffice.To address it, we introduce additional hierarchical pooling mechanism, leading to the *hierarchical pooling encoder* (HPE). Sequence $\mathcal{S}_{\mathbf{T}}$ is first split into chunks of a fixed length $L \in \mathbb{N}_+$: $\mathcal{S}_{\mathbf{T}}^1, \mathcal{S}_{\mathbf{T}}^2, ...$ (the last chunk might be shorter). Topological encoding is applied in parallel to each chunk. The procedure is repeated for the resulting sequence of the topological encodings, which is already shorter by the multiplicative factor of $L$ (potentially with a different $L$ even though here we assume hat $L$ is the same). The total number of repetitions is in practice a small constant $h_{\text{pool}} \in \mathbb{N}$ and leads to the final sequence of length $l = \frac{\text{len}(\mathcal{S}_{\mathbf{T}})}{L^{h_{\text{pool}}}}$, where $\text{len}(\mathcal{S}_{\mathbf{T}})$ is the original length and $l$ as a small constant. If $\text{len}(\mathcal{S}_{\mathbf{T}})$ is small enough, hierarchical pooling is not applied. The resulting $l$-length sequence of latent $d$-dimensional encodings is output as a topological encoding and fed to the temporal encoder defined below. We refer to different tokens of that sequence as *meta-tokens*, $\mathcal{M}$.

### 3.3 Temporal encoder: Compact Associative Memory (CAM) model

**Preliminaries.** The temporal module consists of one or more temporal encoders stacked together that process the meta-tokens, $\mathcal{M}$. It treats the length $l$ of the sequence $\mathcal{M}$ as a batch size $b$. For a given meta-token, denote by $\{\xi^\mu\}_{\mu=1}^M \subseteq \mathbb{R}^d$ its corresponding latent encodings obtained over time. We will refer to them here as *memory-vectors* (patterns). We obtain their latent embeddings: *queries*, *keys* and *values* via learnable linear transformations $\mathbf{W}_Q, \mathbf{W}_K \in \mathbb{R}^{N \times d}, \mathbf{W}_V \in \mathbb{R}^{d \times d}$ as follows:

$$\mathbf{q}^\mu = \mathbf{W}_Q \xi^\mu, \quad \mathbf{k}^\mu = \mathbf{W}_K \xi^\mu, \quad \mathbf{v}^\mu = \mathbf{W}_V \xi^\mu \tag{2}$$

Take a kernel $\mathrm{K} : \mathbb{R}^N \times \mathbb{R}^N \to \mathbb{R}$, and its linearization: $\mathrm{K}(\mathbf{x}, \mathbf{y}) = \mathbb{E}[\phi(\mathbf{x})^\top \phi(\mathbf{y})]$ for some (randomized) $\phi : \mathbb{R}^N \to \mathbb{R}^r$. We refer to $\phi(\mathbf{x}), \phi(\mathbf{y})$ as *random feature* (RF) vectors and define a *hidden state* encapsulating memory of the system of first $t$ patterns as:

$$\begin{cases} \mathbf{N}_t = \sum_{\mu=1}^t \lambda_t(\mu) \phi(\mathbf{k}^\mu)(\mathbf{v}^\mu)^\top \in \mathbb{R}^{r \times d}, \\ \Psi_t = \sum_{\mu=1}^t \lambda_t(\mu) \phi(\mathbf{k}^\mu) \in \mathbb{R}^r \end{cases} \tag{3}$$

A *discount-function* $\lambda_t : \mathbb{R} \to \mathbb{R}$ is applied to deprioritize older patterns.

### 3.3.1 Updating hidden states and the outputs of the temporal module

Note that temporal encoder's hidden state $\mathbf{h}_{\text{Mne}}(t) = (\mathbf{N}_t, \Psi_t)$ is of size **independent** from the number of its implicitly stored patterns $t$. When patterns are added, $\mathbf{h}_{\text{Mne}}(t)$ needs to be efficiently updated on-the-fly. It is easy to see that this can be done for the *exponential discount* strategy: $\lambda_t(\mu) = \exp(-\tau(t - \mu))$ with $\tau \geq 0$ ($\tau = 0$ turns off discounting). We have the following:

$$\mathbf{N}_{t+1} = \exp(-\tau) \cdot \mathbf{N}_t + \phi(\mathbf{k}^{t+1})(\mathbf{v}^{t+1})^\top,$$
$$\Psi_{t+1} = \exp(-\tau) \cdot \Psi_t + \phi(\mathbf{k}^{t+1}) \tag{4}$$

With the definition of the temporal encoder's hidden state, we can now explain how it acts on the input vectors $\xi \in \mathbb{R}^d$. New vector $\xi' = \xi + \Delta\xi$ is obtained as follows, where $\mathbf{q} = \mathbf{W}_Q \xi$:

$$\Delta\xi = \frac{\mathbf{N}_t^\top \phi(\mathbf{q})}{\phi(\mathbf{q})^\top \Psi_t} = \sum_{\mu=1}^t \frac{\lambda_t(\mu) \phi(\mathbf{q})^\top \phi(\mathbf{k}^\mu)}{\sum_{\text{ß}=1}^t \lambda_t(i) \phi(\mathbf{q})^\top \phi(\mathbf{k}^i)} \mathbf{v}^\mu \tag{5}$$

Vector $\xi'$ can be computed in time $O_M(1)$ and is given as a convex combination of value vectors $\mathbf{v}^\mu$, with coefficients proportional to approximated kernel values, but modulated by the discount-function.

For $\mathbf{W}_Q = \mathbf{W}_K = \mathbf{W}_V = \mathbf{I}_d$, $\lambda_t \equiv 1$ and with exact kernel values $\mathrm{K}(\mathbf{q}, \mathbf{k}^i)$ in Eq. 5 (rather than their approximated versions $\phi(\mathbf{q})^\top \phi(\mathbf{k}^i)$), dynamical systems defined by Eq. 5 become effectively Hopfield networks and, as energy-based models with energies given as $E(\xi; \{\xi^\mu\}_{\mu=1}^t) = -\sum_{\mu=1}^t \mathrm{K}(\xi, \xi^\mu)$, retrieve memory-vectors upon energy-minimization-driven convergence [56]. The retrieval quality depends on the kernel, with the softmax-kernel $\mathrm{K}(\mathbf{x}, \mathbf{y}) \stackrel{\text{def}}{=} \exp(\mathbf{x}^\top \mathbf{y})$ providing particularly strong theoretical results. For arbitrary $\mathbf{W}_Q, \mathbf{W}_K, \mathbf{W}_V$, but still with $\lambda_t \equiv 1$ and exact kernel values, Eq. 5 turns into regular Transformers' attention. Upon replacement of the exact kernel values with the approximate ones, Performer model is recovered.

We think about Eq. 5 as a *generalized* (since it uses $\lambda_t$) *compact* (since it provides efficient hidden state and input update-rules via linearized kernels from Performers) *associative **memory*** model (CAM) (as opposed to the regular associate memory model from Hopfield networks).

In Mnemosyne, we choose softmax-kernel as K, since it is a default choice for Transformers. We observed that the FAVOR++ mechanism defining $\phi$ from [44], and denoted by us as $\phi_{F++}$, provides the most robust performance, but since it is harder to implement in the temporal encoder setting (see: discussion in Sec. A.1), in practice we apply the so-called *hyperbolic cosine random features* from [18], performing very similarly, as $\phi_{F++}$. More details including in particular exact definitions and ablation studies over different random feature mechanisms can be found in Sec. A.1, B.3.

### 3.4 Putting it all together

Details of the hierachical pooling encoder (HPE) and the compact associative memory (CAM) modules are depicted in Fig. 1. With all the components of Mnemosyne described, we present the complete design of Mnemosyne for two application modes: **coordinate-wise** and **tensor-wise**, shown in Fig. 2. In the coordinate-wise application, the enriched gradient input $\mathbf{r}_i$ (see: Sec. 3.2) of every parameter $x_i$ is processed separately in parallel by a CAM module followed by an MLP layer to produce the update $\Delta x_i$. This makes the optimizer design simple and allows it to learn optimization from small-scale problems since each parameter input serves as a separate training example. In this setting, CAM stores a fix-sized memory state for each parameter thereby making the optimizer memory state scale linearly with the number of parameters. In the tensor-wise application, every tensor $\mathcal{S}_\mathbf{T}$ in the parameter-tree is processed as a whole by an HPE to produce meta-tokens $\mathcal{M}$ for CAM. Now the CAM memory state becomes very compact because it stores a state for each of the (small number of) meta-tokens rather than every token in the original input sequence. CAM output $\mathcal{L}$ has the same shape as $\mathcal{M}$. It is transformed into a fix-sized encoding $\mathbf{e}$ by a spatial attention encoder (SPE). This encoding is broadcast to all the input tokens of the tensor via concatenations with vectors $\mathbf{r}$. Each resulting vector $\mathbf{r}' = \mathbf{r} \odot \mathbf{e}$ is processed by a single MLP-layer to get the update tensor $\Delta \mathbf{T}$.

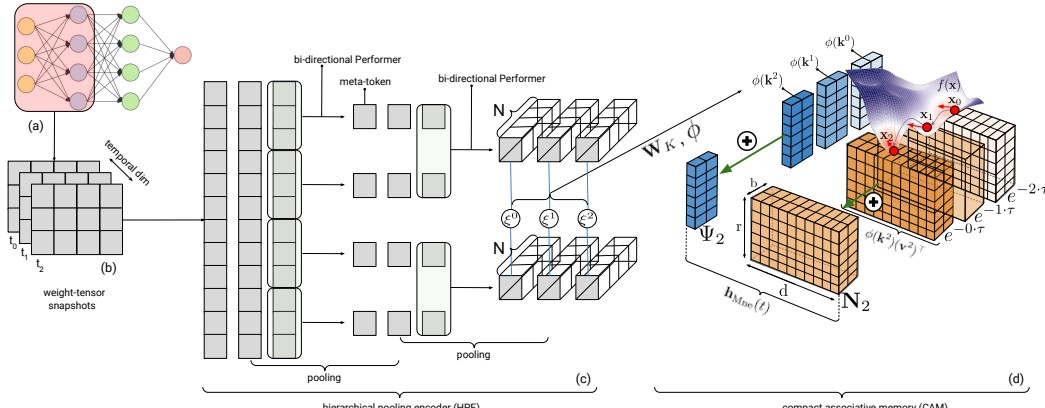

Figure 1: Pictorial description of the hierarchical pooling encoding (HPE) and compact associative memory (CAM) in Mnemosyne on the example of modifying a single weight-tensor of a given NN. Consider training a single $3 \times 4$ weight tensor of a toy feedforward fully connected NN ((a)). Three snapshots of this tensor ((b)) represent its consecutive instantiations in the optimization process. HPE ((c)) acts as follows. Each tensor is vectorized and chunked into sub-sequences that are spatially encoded by the bi-directional Performers. Presented pooling mechanism consists of two layers. This results in the input tensors $\xi^0, \xi^1, \xi^2$ to CAM. Here the first dimension (the final number of meta-tokens) serves as a batch one ($b = 2$) and $N = 4$ (see: notation from Sec. 3.3). Those tensors are first linearly mapped via $\mathbf{W}$ matrices to keys (shown above) and queries and then non-linearly transformed via the $\phi$-mapping. The transformed variants are leveraged by the associative memory.

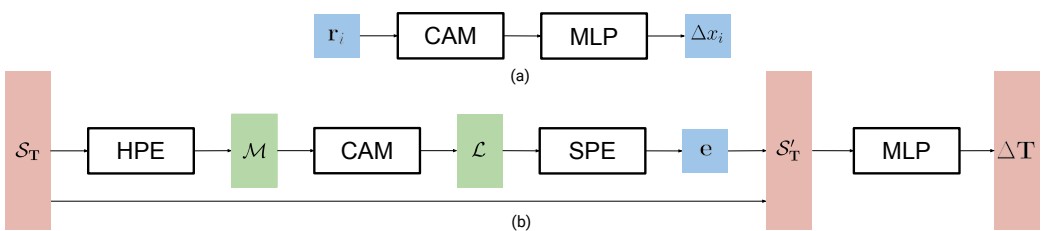

Figure 2: Two modes of Mnemosyne application: (a) **coordinate-wise** and (b) **tensor-wise**.

## 4 The theory of Mnemosyne's compact associative memory

In this section, we analyze Mnemosyne's compact associative memory (CAM) from the theoretical point of view. We show that, as its regular non-compact counterpart, it is capable of storing patterns, but (as opposed to the former) in the implicit manner. We start by providing a concise introduction to associative memory models as "analytical" (rather than combinatorial) energy-based nearest-neighbor search systems. We then present our main result (Theorem 4.3) stating that "on average" CAMs can restore exponentially many (in space dimensionality) number of patterns, provided that they are spread "well enough". To the best of our knowledge, this is the first result of this type.

### 4.1 Regular exponential associative memory

As in several other papers providing a theoretical analysis of the associative memory, we consider feature vectors taken from the set $\{-1, +1\}^N$. We denote by $\{\xi^\mu\}_{\mu=1}^M$ the set of all the memory-vectors to be stored. For the given input $\xi$, in the regular exponential associative memory model, the energy of the system is defined as:

$$E_{\text{reg}}(\xi; \xi^1, ...\xi^M) = -\sum_{\mu=1}^M \exp(\xi^\top \xi^\mu). \tag{6}$$

The dynamical system defining the interactions with the associative memory and whose goal is to retrieve the relevant memory for the given input vector $\sigma \in \{-1, +1\}^N$ (its nearest neighbor in $\{\xi\}_{\mu=1}^M$) has the following form: $\sigma \to T_{i_1}(\sigma) \to T_{i_2}(T_{i_1}(\sigma)) \to ...$, for the initial point $\sigma$, where $i_1, i_2, ...$ are chosen independently at random and $T_j : \{-1, +1\}^N \to \{-1, +1\}^N$ only updates the jth entry of its input as follows (for $\xi[j; x]$ denoting vector $\xi$, but with its jth dimension equal to $x$):

$$T_j(\xi)[j] = \text{sgn}[E_{\text{reg}}(\xi[j; -1]; \xi^1, ...\xi^M) - E_{\text{reg}}(\xi[j; 1]; \xi^1, ...\xi^M)] \tag{7}$$

Thus at every step, a random dimension of the input vector is chosen and its value is flipped if that operation decreases the energy of the system.

**Definition 4.1** (the capacity of the associative models). We say that the model described above stores memories $\{\xi^\mu\}_{\mu=1}^M$ if there exists $\rho \in (0, \frac{1}{2})$ such that $T_j(\widehat{\xi}^\mu)[j] = \xi_j^\mu$ for any $j$ and $\widehat{\xi}^\mu$ taken from the Hamming ball $\mathcal{B}(\xi^\mu, \rho N)$ centered in $\xi^\mu$ and of Hamming radius $\rho N$.

It was proven in [21] that if the memories are chosen uniformly and independently at random from $\{-1, +1\}^N$, then with probability approaching 1 as $N \to \infty$ the model stores all the memories as long as the memory-set is not too large. Most importantly, the upper bound on the memories-set size is exponential in $N$.

*Remark* 4.2. Despite the exponential capacity of the model, all its memories need to be explicitly stored (to compute the energy-function) and compute time is proportional to their number. Thus for large number of memories, the space and time complexity makes the retrieval infeasible in practice.

### 4.2 Mnemosyne's Compact Associative Memory (CAM)

We denote by $\omega_1, \omega_2, ...\omega_r$ samples chosen independently from the multivariate Gaussian distribution $\mathcal{N}(0, \mathbf{I}_N)$ and define the energy of the system as follows (see: Sec. 3.3.1, A.1):

$$E_{\text{rand}}(\xi; \xi^1, ...\xi^M) = \phi_{F++}(\xi)^\top \mathbf{M}(\xi^1, ..., \xi^M), \tag{8}$$

where $\mathbf{M}$ is given as: $\mathbf{M}(\xi^1, ..., \xi^M) = -\sum_{\mu=1}^M \phi_{F++}(\xi^\mu)$. We refer to the number of random projection vectors $\omega$ as the number of *random features* (RFs).

Equipped with this new energy function, we define the corresponding dynamical system in the same way as for the regular associative memory model. Calculating energy $E_{\text{rand}}$ can be now

done efficiently in time $O(r)$, once vector $\mathbf{M}$ is computed (thus independently from the number of implicitly stored memories). In the online/streaming setting, when the memories come one by one, updating vector $\mathbf{M}$ can be done in time $O(Nr)$ per query.

### 4.3 The capacity of Mnemosyne's memory

We are ready to present our main theoretical result. We assume the setting from Sec. 4.2. The extended version of this result, providing in addition concentration results, is given in Sec. A.2.

**Theorem 4.3** (storage of compact associative memories). *Denote by $\xi^1, ..., \xi^M \in \{-1, +1\}^N$ the memory-vectors. Assume that the Hamming distance between any two memory-vectors is at least $\tau N$ for some $\tau > 0$. Take some $0 < \rho < \frac{\tau}{2}$. Then the following is true for any memory-vector $\xi^l$ for $l = 1, ..., \mu$ and any input $\widehat{\xi}^l \in \mathcal{B}(\xi^l, \rho N)$ as long as $M \leq \exp(2N(\tau - 2\rho))\frac{1-e^{-2}}{2e^2}$: the expected change of the energy of the compact associative memory system $\Delta(E_{\mathrm{rand}})$ associated with flipping the value of the dimension of $\widehat{\xi}^l$ is positive if that operation increases the distance from its close neighbor $\xi^l$ and is negative otherwise.*

*Remark* 4.4. Theorem 4.3 says that the expected value of the change of the energy of the system has the correct sign even if the number of stored patterns is exponential in their dimensionality, provided that the patterns are well separated. By the analogous argument as the one from the proof of Theorem 4.3 in Sec. A.2, a similar statement for the method applying other RF-mechanism for softmax-kernel estimation can be derived. However $\phi_{F++}$ provides **the smallest** variance among all competitors as the most accurate currently known mechanism for the unbiased softmax-kernel approximation.

## 5 Experiments

This section is organized as follows. Sec. 5.1 is a warm-up, where we provide initial comparison of Mnemosyne with several hand-designed optimizers and an LSTM-based learned optimizer baseline, on the tasks of training smaller NN architectures. Our results show that Mnemosyne consistently outperforms other variants and that popular $\mathrm{Adam}$ optimizer [34] is the second best. Note also that $\mathrm{Adam}$ is an optimizer of choice for training large Transformer models. Therefore in the following sections, we focus on the detailed comparison of Mnemosyne with $\mathrm{Adam}$ for larger architectures.

In Sec. 5.2, we test the **coordinate-wise Mnemosyne** for ViT fine-tuning and soft prompt-tuning 11B+ T5XXL models. Optimizer's memory scales linearly with the NN size for the coordinate-wise variant and $\mathrm{Adam}$. However, depending on the size and number of temporal attention layers in the coordinate-wise Mnemosyne, the linear multiplicative constant can be prohibitively large. This restriction is alleviated by the tensor-wise variant. In Sec. 5.3, we present the results for the **tensor-wise Mnemosyne** on BERT MLM pre-training and ViT fine-tuning. The tensor-wise Mnemosyne's memory state scales with the number of tensors instead of the number of model parameters. However, this variant requires meta-training with larger tensors for best generalization results since now each tensor serves as a single training example instead of each parameter which was the case for the coordinate-wise variant. To take advantage of the efficiency of the tensor-wise and the efficacy of the coordinate-wise variant, we present **Super-Mnemosyne**, in Sec. 5.4, that merges them.

All Mnemosyne variants are meta-trained on small scale MLP and VIT training tasks for a short horizon of 100 steps. More detailed description of the meta-training set-up is given in the Appendix (Sec: B) along with the details of the experiments in this section. Additional results including: (a) comparison of the CAM mechanism with regular attention (Sec. B.2), (b) studies over different RF-mechanisms (Sec. B.3), ablations over: (c) different discount factors and different number of random features for the CAM mechanism (Sec. B.5) and (d) different depths of the temporal modules of Mnemosyne (Sec. B.6) are also given in the Appendix.

We have finalized the design choices for Mnemosyne optimizer used throughout this section based on the extensive theoretical analysis in previous sections and various ablation studies detailed in the Appendix. In particular:

1. The choice of the random feature (RF) mechanism is motivated by strong theoretical guarantees for the capacity of the corresponding associative memory. Other RF mechanisms that do not admit our theoretical analysis clearly underperform in the ablation results.

2. Our default hyperbolic cosine RF mechanism is motivated by the provably reduced estimation variance as explained previously.

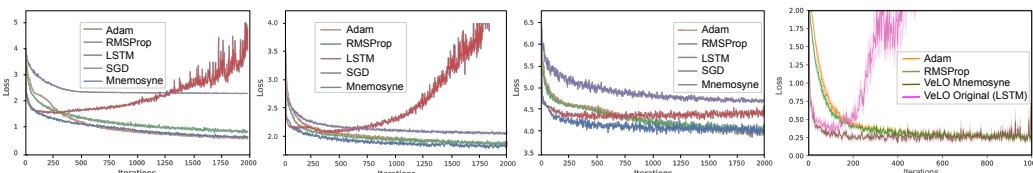

Figure 3: **First three plots:** Test loss curves for training ViTs with Mnemosyne on: MNIST, CIFAR10 and CIFAR100. Minimal feature engineering is applied. **Last plot**: Test loss curves for training an MLP on MNIST. Both Mnemosyne and LSTM are used within $\mathcal{V}e\mathcal{LO}$ architecture using sophisticated feature engineering.

3. We choose hyperbolic cosine RFs as they are shown more robust than regular positive RFs in ablation.

4. Our use of exponential discount strategy (EDS) to smoothly discount the past is theoretically motivated previously. We show empirical verification of EDS in the appendix and choose the discount factor based on ablation results.

## 5.1   Warm-up: Mnemosyne vs other optimizers for smaller NN architectures

We start by comparing coordinate-wise Mnemosyne with standard optimizers: $\mathrm{Adam}$ [34], $\mathrm{RMSProp}$ [78], $\mathrm{SGD}$ [33] as well as popular learnable optimizers using LSTMs [2]. All optimizers were tested on the task of training Vision Transformer (ViT) [23] architectures not seen during meta-training. Considered ViT has 3 attention and MLP layers of size 16 and 2 heads. Loss curves for image classification with ViTs on different datasets (MNIST, CIFAR10, CIFAR100) are shown as first three plots in Fig. 3. Minimal feature engineering techniques were applied. In the last plot of Fig. 3, we inserted Mnemosyne into $\mathcal{V}e\mathcal{LO}$ optimizer [48] which originally used LSTMs. This learnable optimizer applies sophisticated feature engineering to mitigate the problem of catastrophic forgetting of the LSTM-cells. Note that we did not replicate their large-scale meta-training set-up, only the optimizer architecture and features. Mnemosyne outperforms its counterparts in all these scenarios. In particular, it successfully trains attention-based architectures on unseen tasks.

Now we proceed with the comparison of Mnemosyne and $\mathrm{Adam}$ with different learning rates on larger Transformers. We emphasize that we do not conduct any hyperparameter tuning for Mnemosyne.

## 5.2   Coordinate-wise Mnemosyne results

**ViT last-layer fine-tuning:** We tested a coordinate-wise Mnemosyne with 2 temporal encoders on ViT fine-tuning [23]. Due to memory constraints, we only trained the last layer of ViT by freezing other parameters of the model. The results on three different datasets: $\mathrm{imagenet}2012$, $\mathrm{places}365$ and $\mathrm{caltech\text{-}birds\text{-}2011}$ for ViT-H(32) (36 layers, 16 heads, $32 \times 32$ patch shape) are presented in Fig. 4. Additional results, including ablations with varying architecture sizes and other datasets, are shown in Appendix Sec: B.8. We conclude that Mnemosyne (without any hyperparameter tuning) matches the top-performing $\mathrm{Adam}$ variant. Note that $\mathrm{Adam}$ is very sensitive to the choice of the learning rate $\mathrm{lr}$.

In Table 1 we compare coordinate-wise Mnemosyne with two other classes of learnable optimizers: a) Those using S4 memory units [27] based on state space machines as an alternative to LSTMs and Transformers b) The Lion family of learnable optimizers [14] using a different approach of symbolic programming. In the case of ViT-L architecture, we skip S4 due to poor performance on the previous tasks. Mnemosyne is clearly the best among all three variants. In contrast to Lion, Mnemosyne does not require any hyperparameter tuning.

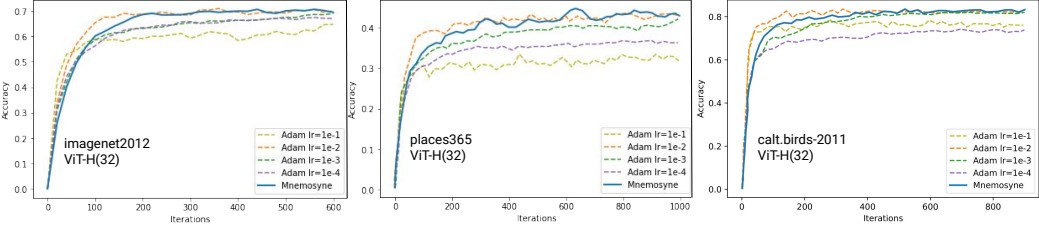

Figure 4: Fine-tuning the last layer of ViT-H(32) on different datasets with coordinate-wise Mnemosyne.

| Method | ViT-H (ps=14) | ViT-H (ps=8) | ViT-L (ps=32) |
|---|---|---|---|
| Mnemosyne | 72.3 | 81.4 | 74.0 |
| S4 | 7.5 | 35.0 | - |
| Lion lr=1e-1 | 67.2 | 76.4 | 66.8 |
| Lion lr=1e-2 | 68.3 | 80.0 | 70.0 |
| Lion lr=2e-3 | 70.0 | 80.0 | 71.4 |
| Lion lr=1e-3 | 72.3 | 81.2 | 71.4 |
| Lion lr=1e-4 | 70.0 | 81.0 | 71.4 |

Table 1: Comparison of coordinate-wise Mnemosyne with Lion and S4-based learned optimizer. Accuracy after fine-tuning the last layer of different ViT architectures and different patch sizes (ps) on Imagenet2012 dataset for 500 iterations.

**ViT multi-layer fine-tuning:** The *light Mnemosyne* variant with a single temporal encoder can scale up to fine-tune 2 Transformer layers along with the last layer. We show the result for fine-tuning ViT-B(16) on $\mathrm{Cifar100}$ in Fig. 5. The non-Mnemosyne layers are trained using an untuned $\mathrm{Adam}$ and the Adam variants for comparison fine-tune the full model. We observe that full-model finetuning is a hard task where all but one $\mathrm{Adam}$ variant fail to reach the optimal accuracy. Here, even by training a part of the network with Mnemosyne, we achieve the best accuracy. Also note that the best $\mathrm{Adam}$

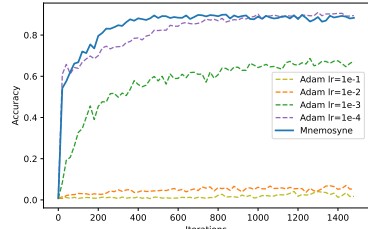

Figure 5: Fine-tuning ViT-B(16).

learning rate in this experiment ($1e^{-4}$) was performing poorly for the previous one. We see this in subsequent results as well.

**Soft prompt-tuning T5XXL:** We tested coordinate-wise Mnemosyne for soft prompt-tuning 11B+ T5XXL Transformers [55] on the SuperGLUE task [38]. This method was introduced as a scalable alternative to fine-tuning pre-trained models for several downstream tasks, with learnable prompts injected into Transformer layers and modulating its behaviour. The trainable soft-prompt contains 12288 parameters. Mnemosyne outperforms all $\mathrm{Adam}$ variants (see: Fig. 6).

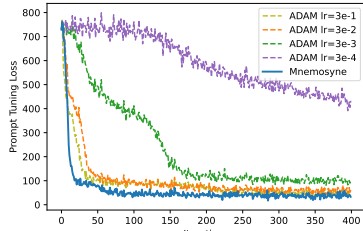

Figure 6: Soft prompt-tuning T5XXL.

### 5.3 Tensor-wise Mnemosyne results

In this section, we evaluate the memory-efficient tensor-wise Mnemosyne.

**BERT NLP Transformer pre-training:** We showcase the ability of tensor-wise Mnemosyne to pre-train a BERT-base text Transformer [22] on the standard masked language modeling (MLM) task.

Coordinate-wise Mnemosyne was not applicable here due to the memory footprint and thus it became a practical test for the memory efficient tensor-wise variant. With that variant, we were able to train **86M parameters** of Bert-base model thereby showcasing the scalability of tensor-wise Mnemosyne. We compare Mnemosyne with different variants of $\mathrm{Adam}$ in Fig 7. We see that Mnemosyne matches the performance of the best $\mathrm{Adam}$ variant *even though it was never exposed to the MLM task during meta-training*. Several $\mathrm{Adam}$ variants get stuck in a local optima, worse than that found using Mnemosyne.

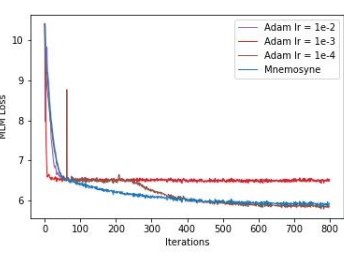

Figure 7: MLM task with BERT.

**ViT multi-layer fine-tuning:** Now we show the performance of tensor-wise Mnemosyne on fine-tuning ViTs. Although tensor-wise Mnemosyne can scale up to the full ViT model, here we freeze large tensors in the Transformer layers and fine-tune the rest. Training large tensors with tensor-wise Mnemosyne requires data and compute intensive large scale meta-training (to ensure that spatial encoders learn how to encode long sequences well). This will be the focus of the future work. For this experiment, we consider the ViT-H(32) model and three datasets:

Cifar100, places365, imagenet in Fig. 8. Mnemosyne shows equivalent or better performance as compared to the optimal Adam.

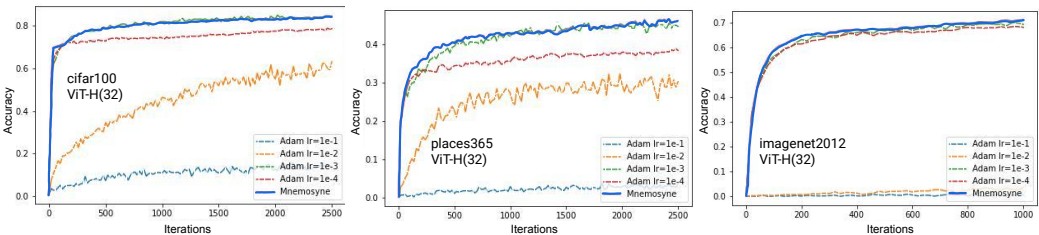

Figure 8: Fine-tuning multi-layer ViT-H(32) on different datasets with tensor-wise Mnemosyne.

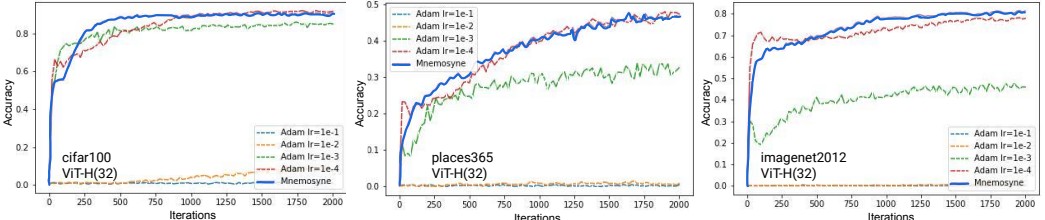

Figure 9: Fine-tuning **full** ViT-B(16) on different datasets with Super-Mnemosyne.

### 5.4 Super-Mnemosyne: combining coordinate- and tensor-wise strategies

In this last set of ViT-experiments, we combine tensor-wise and coordinate-wise light Mnemosyne. Since applying the former on large tensors requires more intense meta-training which is out of the scope of this paper, we decided to optimize the largest tensors with the coordinate-wise and others with the tensor-wise Mnemosyne (to keep meta-training simple). This combination turned out to minimize total memory usage. As we see in Fig. 9, Mnemosyne outperforms optimal Adam variants.

## 6 Broader impacts & limitations

We believe that Mnemosyne opens a research on attention-based optimizers for general-purpose optimization. Our system should be used responsibly due to the potential for misuse, significant societal impact, and carbon footprint of Transformers [3, 10, 70]. In the future we want to analyze in more depth the impact on more complex meta-training strategies on the quality of learned optimizers.

## 7 Conclusion

We proposed a new class of learnable optimizers applying efficient spatio-temporal attention, called Mnemosyne. We show that they outperform their LSTM-based counterparts and can be successfully used to fine/soft prompt-tune and pre-train large Transformer models, matching optimal hard-coded variants without any hyper-parameter tuning, often producing top performing models.

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

# A Proofs

## A.1 Kernels & their linearizations for temporal encoders in Mnemosyne

We tested different transformations $\phi$ and discovered that those leading to most accurate approximation of the softmax-kernel lead to most effective memory mechanisms for Mnemosyne's temporal encoders (see: Sec. 3.3). Our starting variant is the so-called *FAVOR+* mechansim from [18], given as follows for $\Gamma(\mathbf{z}, r) \overset{\text{def}}{=} \frac{1}{\sqrt{r}} \exp(-\frac{\|\mathbf{z}\|^2}{2})$ and $\omega_1, ..., \omega_r \sim \mathcal{N}(0, \mathbf{I}_N)$:

$$\phi_{F+}(\mathbf{z}) = \Gamma(\mathbf{z}, r) \left( \exp(\omega_1^\top \mathbf{z}), ..., \exp(\omega_r^\top \mathbf{z}) \right)^\top \tag{9}$$

Random vectors $\omega_1, ..., \omega_r$ form a block-orthogonal ensemble (see: [18]). We applied also its improvement relying on the so-called *hyperbolic cosine random features*, where $\prod$ is the concatenation operator:

$$\phi_{HF+}(\mathbf{z}) = \Gamma(\mathbf{z}, r) \prod_{i=1}^{\frac{r}{2}} (\exp(\omega_i^\top \mathbf{z}), \exp(-\omega_i^\top \mathbf{z}))^\top \tag{10}$$

Both randomized transformations provide **unbiased** estimation of the softmax-kernel, yet the latter one (that can be cast as modified $\phi_{F+}$ via the antithetic Monte Carlo trick) has provably lower approximation variance.

### A.1.1 The curious case of linearization with bounded features

The last variant for the efficient estimation of the softmax-kernel we applied, is a very recent mechanism *FAVOR++* from [44], given as:

$$\phi_{F++}(\mathbf{z}) = \frac{D}{\sqrt{r}} \prod_{i=1}^{r} \exp(-\widehat{A}\|\omega_i\|_2^2 + B\omega_i^\top \mathbf{z} + C\|\mathbf{z}\|^2)^\top,$$

where we have: $\widehat{A} = -A$, $B = \sqrt{1 + 4\widehat{A}}$, $C = -\frac{1}{2}$, $D = (1 + 4\widehat{A})^{\frac{N}{4}}$, $A = 1 - \frac{1}{\rho}$ and $\rho \in (0, 1)$ is a free parameter. As opposed to the previous variants, mechanism $\phi_{F++}(\mathbf{z})$ provides an estimation via **bounded** random variables (since $\widehat{A} > 0$), leading to stronger concentration results (beyond second moment) and still unbiased approximation.

The optimal choice of $\rho$ depends on the kernel inputs. The formula for $\rho$ optimizing the variance of the kernel matrix estimation $\mathcal{K} = [\mathrm{K}(\mathbf{q}^i, \mathbf{k}^j)]_{i,j=1,...,M}$ induced by the softmax-kernel K (in the bi-directional case) is not tractable. However choosing $\rho$ by optimizing certain derivative of the variance-objective was showed to work well in several applications [44]:

$$\rho^* = \frac{\sqrt{(2\gamma + N)^2 + 8N\gamma} - 2\gamma - N}{4\gamma} \tag{11}$$

for $\gamma = \frac{1}{M^2} \sum_{i=1}^{M} \sum_{j=1}^{M} \|\mathbf{q}^i + \mathbf{k}^j\|^2$. Since $\gamma$ can be rewritten as: $\gamma = \frac{1}{M^2}(\sum_{i=1}^{M} \|\mathbf{q}^i\|_2^2 + \sum_{j=1}^{M} \|\mathbf{k}^j\|_2^2 + 2\mathbf{q}^\top \mathbf{k})$ for $\mathbf{q} = \sum_{i=1}^{M} \mathbf{q}^i$ and $\mathbf{k} = \sum_{j=1}^{M} \mathbf{k}^j$, computing $\rho^*$ in the bi-directional setting can be clearly done in time linear in $M$ as a **one-time** procedure. Then the computation of $\mathbf{h}_{\mathrm{Mne}}(M)$ follows. Compute-time per memory-vector remains $O_M(1)$.

However Mnemosyne's temporal encoder applied uni-directional attention. In the uni-directional case, we have: $\gamma_t = \frac{1}{t} \sum_{j=1}^{t} \|\mathbf{q}^t + \mathbf{k}^j\|^2 = \frac{1}{t}(\|\mathbf{q}^t\|_2^2 + \sum_{j=1}^{t} \|\mathbf{k}^j\|_2^2 + 2(\mathbf{q}^t)^\top \mathbf{k}(t))$, where $\mathbf{k}(t) = \sum_{j=1}^{t} \mathbf{k}^j$ for $t = 1, ..., M$. Instead of one $\gamma$, we now have $M$ values $\gamma_t$ since not all memories are known at once. We can still achieve $O_1(M)$ compute-time per $\gamma_t$, repeating the trick from the bi-directional case, but that would need to be followed by the re-computation of $\phi(\mathbf{k}^\mu)$ (with new $\rho$-parameter) for $\mu = 1, ..., t$ which of course is not possible since vectors $\{\mathbf{k}\}_{\mu=1}^{t}$ are not explicitly stored (and for a good reason - computational benefits), see: Eq. 3.

**Thickening Mnemosyne's memory:** To obtain efficient uni-directional Mnemosyne's memory cell also for the $\phi_{F++}$-mechanism, we propose to "thicken" in that setting the hidden state from Eq. 3, replacing $\mathbf{h}_{\mathrm{Mne}}(t) = (\mathbf{N}_t, \Psi_t)$ with $\mathbf{H}_{\mathrm{Mne}}(t) = (\{\mathbf{N}_t^\rho\}_{\rho \in \Omega}, \{\Psi_t^\rho\}_{\rho \in \Omega}, \Sigma_t, \Lambda_t)$, where we have: $\Sigma_t = \sum_{j=1}^{t} \mathbf{k}^j$, $\Lambda_t = \sum_{j=1}^{t} \|\mathbf{k}^j\|_2^2$ and furthermore: $\mathbf{N}_t^\rho$, $\Psi_t^\rho$ correspond to versions of $\mathbf{N}_t$ and $\Psi_t$

respectively, using parameter $\rho$ to define mapping $\phi$. The set $\Omega$ is obtained by discretizing interval $(0,1)$ into a fixed number of chunks $c$ (and effectively quantizes $\rho \in (0,1)$). The strategy is now clear: when the new pattern comes, we first update the entire thickened state, and then compute $\rho^*$. We finalize by finding $\rho \in \Omega$ closest to $\rho^*$ to transform an input and using for that the "slice" of the hidden state corresponding to $\rho$. We see that all these operations can be made efficiently with only $c$-multiplicative term (**independent** from the number of patterns $M$) in space and time complexity.

FAVOR++ mechanism, as FAVOR+, can also be adapted to its hyperbolic cosine variant. In practice FAVOR+ mechanism worked similarly to FAVOR++, yet the proper adaptation of the latter one was important, since (see: Sec. 4), this variant provides strongest theoretical guarantees for the capacity of the entire compact associative memory model.

## A.2 The proof of the extended version of Theorem 4.3

We start by providing an extended version of Theorem 4.3, enriched with the exact formula of the variance of $\Delta(E_{\mathrm{rand}})$. We prove it below. We borrow the notation from Sec. A.1.

**Theorem A.1** (storage of compact associative memories). *Denote by $\xi^1, ..., \xi^M \in \{-1,+1\}^N$ the memory-vectors. Assume that the Hamming distance between any two memory-vectors is at least $\tau N$ for some $\tau > 0$. Take some $0 < \rho < \frac{\tau}{2}$. Then the following is true for any memory-vector $\xi^l$ for $l = 1, ..., \mu$ and any input $\widehat{\xi}^l \in \mathcal{B}(\xi^l, \rho N)$ as long as $M \leq \exp(2N(\tau - 2\rho))\frac{1-e^{-2}}{2e^2}$: the expected change of the energy of the compact associative memory system $\Delta(E_{\mathrm{rand}})$ associated with flipping the value of the dimension of $\widehat{\xi}^l$ is positive if that operation increases the distance from its close neighbor $\xi^l$ and is negative otherwise. Furthermore, the variance of $\Delta(E_{\mathrm{rand}})$ is of the form:*

$$\mathrm{Var}(\Delta(E_{\mathrm{rand}})) = \frac{1}{r}(V_1 + V_2 - 2V_3 - V_4 - V_5 + 2V_6) \tag{12}$$

*where:*

$$V_1 = \sum_{\mu_1,\mu_2 \in \{1,...,M\}} \Psi(\xi^{\mu_1} + \xi^{\mu_2} + 2\widehat{\xi}^l), \quad V_2 = \sum_{\mu_1,\mu_2 \in \{1,...,M\}} \Psi(\xi^{\mu_1} + \xi^{\mu_2} + 2\tilde{\xi}^l)$$

$$V_3 = \sum_{\mu_1,\mu_2 \in \{1,...,M\}} \Psi(\xi^{\mu_1} + \xi^{\mu_2} + \widehat{\xi}^l + \tilde{\xi}^l) \quad V_4 = \sum_{\mu_1,\mu_2 \in \{1,...,M\}} \exp((\xi^{\mu_1})^\top \widehat{\xi}^l) \exp((\xi^{\mu_2})^\top \widehat{\xi}^l)$$

$$V_5 = \sum_{\mu_1,\mu_2 \in \{1,...,M\}} \exp((\xi^{\mu_1})^\top \tilde{\xi}^l) \exp((\xi^{\mu_2})^\top \tilde{\xi}^l) \quad V_6 = \sum_{\mu_1,\mu_2 \in \{1,...,M\}} \exp((\xi^{\mu_1})^\top \widehat{\xi}^l) \exp((\xi^{\mu_2})^\top \tilde{\xi}^l)$$

$$\tag{13}$$

*for $\tilde{\xi}^l$ denoting $\widehat{\xi}^l$ with one of its dimensions flipped and:*

$$\Psi(\mathbf{x}) \overset{\mathrm{def}}{=} D^4 \exp(-2N)(1 + 8\widehat{A})^{-\frac{N}{2}} \exp\left(\frac{B^2}{2(1 - 8\widehat{A})}\|\mathbf{x}\|^2\right) \tag{14}$$

*Proof.* Take a memory $\xi^l \in \{-1,+1\}^N$ and an input $\widehat{\xi}^l \in \mathcal{B}(\xi^l, \rho N)$. Denote by $\mathrm{neg}(\widehat{\xi}^l, i)$ a vector obtained from $\widehat{\xi}^l$ by replacing $\widehat{\xi}^l(i)$ with $-\widehat{\xi}^l(i)$. Let us study the change of the energy of the system as we flip the value of the $i$th dimension of the input $\widehat{\xi}^l$ since the sign of this change solely determines the update that will be made. We have the following:

$$\Delta(E_{\mathrm{rand}}) = E(\mathrm{neg}(\widehat{\xi}^l, i); \xi^1, ..., \xi^M) - E(\widehat{\xi}^l; \xi^1, ..., \xi^M) = E_{\mathrm{signal}} + E_{\mathrm{noise}}, \tag{15}$$

where:

$$E_{\mathrm{signal}} = \frac{1}{r}\sum_{k=1}^{r}(W_k^l - Z_k^l), \tag{16}$$

$$E_{\mathrm{noise}} = \frac{1}{r}\sum_{k=1}^{r}\sum_{\mu \in \{1,...,M\}\setminus\{l\}}(W_k^\mu - Z_k^\mu), \tag{17}$$

and furthermore: $W_k^i = a_k^i b_k$, $Z_k^i = a_k^i c_k$ for:

$$a_k^i = D \exp(-\frac{N}{2}) \exp(B\omega_k^\top \xi^i - \widehat{A}\|\omega_k\|_2^2),$$

$$b_k = D \exp(-\frac{N}{2}) \exp(B\omega_k^\top \widehat{\xi}^l - \widehat{A}\|\omega_k\|_2^2),$$

$$c_k = D \exp(-\frac{N}{2}) \exp(B\omega_k^\top \operatorname{neg}(\widehat{\xi}^l, i) - \widehat{A}\|\omega_k\|_2^2).$$

(18)

If $\omega_1, ..., \omega_r \sim \mathcal{N}(0, \mathbf{I}_N)$ then, from the fact that $E_{\mathrm{rand}}$ is the unbiased estimation of $E_{\mathrm{reg}}$, we get:

$$\mathbb{E}[X_k] = \exp((\xi^l)^\top \widehat{\xi}^l),$$

$$\mathbb{E}[Y_k] = \exp((\xi^l)^\top \operatorname{neg}(\widehat{\xi}^l, i)),$$

$$\mathbb{E}[W_k^\mu] = \exp((\xi^\mu)^\top \widehat{\xi}^l),$$

$$\mathbb{E}[Z_k^\mu] = \exp((\xi^\mu)^\top \operatorname{neg}(\widehat{\xi}^l, i)),$$

(19)

This is a direct consequence of the OPRF-mechanism introduced in [44]. Variables: $X_k$, $Y_k$, $W_k^\mu$ and $Z_k^\mu$ for $\mu = 1, ..., M$ are simply unbiased estimators of the softmax-kernel values obtained via applying OPRF-mechanism. Let us now compute the expected change of the energy of the system:

$$\mathbb{E}[\Delta(E_{\mathrm{rand}})] = \mathbb{E}[E_{\mathrm{signal}}] + \mathbb{E}[E_{\mathrm{noise}}],$$

(20)

where:

$$\mathbb{E}[E_{\mathrm{signal}}] = \frac{1}{r} \sum_{k=1}^{r} (\mathbb{E}[X_k] - \mathbb{E}[Y_k]) = \frac{1}{r} \sum_{k=1}^{r} \left( \exp((\xi^l)^\top \widehat{\xi}^l) - \exp((\xi^l)^\top \operatorname{neg}(\widehat{\xi}^l, i)) \right)$$

(21)

and

$$\mathbb{E}[E_{\mathrm{noise}}] = \frac{1}{r} \sum_{k=1}^{r} \sum_{\mu \in \{1,...,M\} \setminus \{l\}} (\mathbb{E}[W_k^\mu] - \mathbb{E}[Z_k^\mu]) =$$

$$\frac{1}{r} \sum_{k=1}^{r} \sum_{\mu \in \{1,...,M\} \setminus \{l\}} \left( \exp((\xi^\mu)^\top \widehat{\xi}^l) - \exp((\xi^\mu)^\top \operatorname{neg}(\widehat{\xi}^l, i)) \right)$$

(22)

We will first upper bound $|\mathbb{E}[E_{\mathrm{noise}}]|$. We have:

$$|\mathbb{E}[E_{\mathrm{noise}}]| \leq \frac{1}{r} \sum_{k=1}^{r} \sum_{\mu \in \{1,...,M\} \setminus \{l\}} \left( \exp((\xi^\mu)^\top \widehat{\xi}^l) + \exp((\xi^\mu)^\top \operatorname{neg}(\widehat{\xi}^l, i)) \right)$$

$$\leq \sum_{k=1}^{r} \sum_{\mu \in \{1,...,M\} \setminus \{l\}} (\exp(N(1 - 2(\tau - \rho))) + \exp(N(1 - 2(\tau - \rho) + \frac{2}{N})))$$

(23)

$$\leq 2M \exp(N(1 - 2(\tau - \rho) + \frac{2}{N}))$$

We will now consider two cases:

**Case 1:** $\widehat{\xi}^l(i) = \xi^l(i)$**:**

In this setting, flipping the value of the ith dimension of the input vector increases its distance from the close neighbor. Therefore in this case we would like the energy change of the system to be positive (so that the flip does not occur). From the Equation 21, we obtain:

$$\mathbb{E}[E_{\mathrm{signal}}] \geq \frac{1}{r} \sum_{k=1}^{r} (\exp(N(1 - 2\rho)) - \exp(N(1 - 2\rho) - 2))) =$$

$$\exp(N(1 - 2\rho))(1 - e^{-2})$$

(24)

Thus we obtain:

$$\mathbb{E}[\Delta(E_{\text{rand}})] \geq \exp(N(1-2\rho))(1-e^{-2}) - 2M\exp(N(1-2(\tau-\rho)+\frac{2}{N})) \quad (25)$$

Therefore, if the following holds:

$$M \leq \exp(2N(\tau-2\rho))\frac{1-e^{-2}}{2e^2}, \quad (26)$$

then $\mathbb{E}[\Delta(E_{\text{rand}})] > 0$.

**Case 2:** $\widehat{\xi}^l(i) = -\xi^l(i)$:

In this setting, flipping the value of the ith dimension of the input vector decreases its distance from the close neighbor. Therefore in this case we would like the energy change of the system to be negative (so that the flip does not occur). From the Equation 21, we obtain:

$$\mathbb{E}[E_{\text{signal}}] \leq \frac{1}{r}\sum_{k=1}^{r}\left(\exp(N(1-2\rho)) - \exp(N(1-2\rho)+2))\right) = \\ \exp(N(1-2\rho))(1-e^2) \quad (27)$$

Thus we obtain:

$$\mathbb{E}[\Delta(E_{\text{rand}})] \leq \exp(N(1-2\rho))(1-e^2) + 2M\exp(N(1-2(\tau-\rho)+\frac{2}{N})) \quad (28)$$

Therefore, if the following holds:

$$M \leq \exp(2N(\tau-2\rho))\frac{e^2-1}{2e^2}, \quad (29)$$

then $\mathbb{E}[\Delta(E_{\text{rand}})] < 0$. Note that the bound from Inequality 26 is stronger than the one from Inequality 29. That completes the proof of the first part of the theorem.

Now we will compute the variance of $\Delta(E_{\text{rand}})$. Denote:

$$Z_k = \sum_{\mu\in\{1,\dots,M\}}(W_k^\mu - Z_k^\mu) \quad (30)$$

Note that if $\omega_1,\dots,\omega_r$ are chosen independently then $Z_k$ for $k=1,\dots,r$ are independent. The following is true:

$$\text{Var}(\Delta(E_{\text{rand}})) = \text{Var}(E_{\text{signal}} + E_{\text{noise}}) = \text{Var}\left(\frac{1}{r}\sum_{k=1}^{r}\sum_{\mu\in\{1,\dots,M\}}(W_k^\mu - Z_k^\mu)\right)$$

$$= \text{Var}(\frac{1}{r}\sum_{k=1}^{r}Z_k) = \frac{1}{r^2}\sum_{k=1}^{r}\text{Var}(Z_k) = \frac{1}{r^2}\sum_{k=1}^{r}\text{Var}\left(\sum_{\mu\in\{1,\dots,M\}}(W_k^\mu - Z_k^\mu)\right) \quad (31)$$

$$= \frac{1}{r^2}\sum_{k=1}^{r}\left(\mathbb{E}\left[\left(\sum_{\mu\in\{1,\dots,M\}}(W_k^\mu - Z_k^\mu)\right)^2\right] - \left(\mathbb{E}\left[\sum_{\mu\in\{1,\dots,M\}}(W_k^\mu - Z_k^\mu)\right]\right)^2\right)$$

Therefore we have:

$$\text{Var}(\Delta(E_{\text{rand}})) = \frac{1}{r^2} \sum_{k=1}^{r} \left( \sum_{\mu_1,\mu_2 \in \{1,\ldots,M\}} \mathbb{E}[W_k^{\mu_1} W_k^{\mu_2}] + \sum_{\mu_1,\mu_2 \in \{1,\ldots,M\}} \mathbb{E}[Z_k^{\mu_1} Z_k^{\mu_2}] \right)$$

$$- \frac{2}{r^2} \sum_{k=1}^{r} \sum_{\mu_1,\mu_2 \in \{1,\ldots,M\}} \mathbb{E}[W_k^{\mu_1} Z_k^{\mu_2}]$$

$$- \frac{1}{r^2} \sum_{k=1}^{r} \left( \sum_{\mu_1,\mu_2 \in \{1,\ldots,M\}} \mathbb{E}[W_k^{\mu_1}]\mathbb{E}[W_k^{\mu_2}] + \sum_{\mu_1,\mu_2 \in \{1,\ldots,M\}} \mathbb{E}[Z_k^{\mu_1}]\mathbb{E}[Z_k^{\mu_2}] \right) \quad (32)$$

$$- \frac{2}{r^2} \sum_{k=1}^{r} \sum_{\mu_1,\mu_2 \in \{1,\ldots,M\}} \mathbb{E}[W_k^{\mu_1}]\mathbb{E}[Z_k^{\mu_2}]$$

Note that from the fact that our random feature map based estimators are unbiased, we get (as we already noted before in Equation 19 and put here again for Reader's convenience):

$$\mathbb{E}[W_k^{\mu}] = \exp((\xi^{\mu})^{\top} \widehat{\xi}^l),$$
$$\mathbb{E}[Z_k^{\mu}] = \exp((\xi^{\mu})^{\top} \text{neg}(\widehat{\xi}^l, i)), \quad (33)$$

Let us now define:
$$\Psi(\mathbf{x}) = D^4 \exp(-2N) \exp(B\omega^{\top}\mathbf{x} - 4\widehat{A}\|\omega\|_2^2). \quad (34)$$

Note that the following is true:

$$\mathbb{E}[W_k^{\mu_1} W_k^{\mu_2}] = \Psi(\xi^{\mu_1} + \xi^{\mu_2} + 2\widehat{\xi}^l)$$
$$\mathbb{E}[Z_k^{\mu_1} Z_k^{\mu_2}] = \Psi(\xi^{\mu_1} + \xi^{\mu_2} + 2\text{neg}(\widehat{\xi}^l, i)) \quad (35)$$
$$\mathbb{E}[W_k^{\mu_1} Z_k^{\mu_2}] = \Psi(\xi^{\mu_1} + \xi^{\mu_2} + \widehat{\xi}^l + \text{neg}(\widehat{\xi}^l, i))$$

Thus it remains to find closed-form formula for $\Psi(\mathbf{x})$ for any given $\mathbf{x} \in \mathbb{R}^N$.

From the proof of Theorem 3.1 in [44], we get for $A < 0$:

$$\mathbb{E}[\exp(A\|\omega\|^2 + B\omega^{\top}\mathbf{x})] = (1 - 2A)^{-\frac{N}{2}} \exp\left( \frac{B^2}{2(1-2A)} \|\mathbf{x}\|^2 \right) \quad (36)$$

Thus we obtain:

$$\Psi(\mathbf{x}) = D^4 \exp(-2N)(1 + 8\widehat{A})^{-\frac{N}{2}} \exp\left( \frac{B^2}{2(1-8\widehat{A})} \|\mathbf{x}\|^2 \right) \quad (37)$$

Plugging to Equation 32 formulae from Equation 33 and Equation 35 and utilizing Equation 37 for $\Psi$, we obtain the formula for the variance from the statement of the Theorem. $\qquad \square$

## B  Experiment details

### B.1  Warm-up for Mnemosyne and other optimizers: additional results

**Preliminaries:** At each timestep $t$, gradient $\nabla f(\mathbf{x}_t)$ is input to the optimizer. The gradient is pre-processed as proposed in [2]. Coordinate-wise Mnemosyne's using two temporal encoders is applied. The Mnemosyne's memory cell interfaces with the rest of the system similarly to any RNN-cell. Each cell uses exponential discount factor $\tau = 0.1$, $r = 16$ random projections, 16 hidden dimensions and 1 attention head. The memory cell output is fed to a fully connected layer, returning the update to be applied to the NN parameters of the optimizee.

**Meta-training:** We refer to training optimizer's parameters $\theta$ as *meta-training* to distinguish from the optimizee NN training. Mnemosyne's optimizer is meta-trained on MNIST classification task with 3

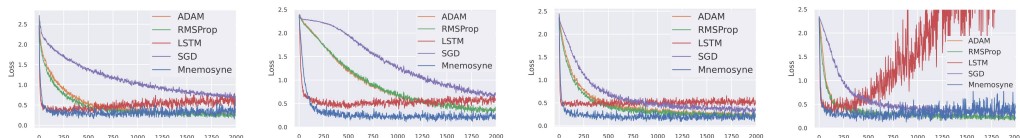

Figure 10: Validation loss curves when training MLP with Mnemosyne compared to other methods for MNIST image classification. Optimization curves for 4 different MLP architectures in this order: (1 layer, 20 hidden dim, sigmoid activation), (2 layers, 20 hidden dim, sigmoid activation), (1 layer, 40 hidden dim, sigmoid activation), (1 layer, 20 hidden dim, relu activation) are shown.

small MLP and 3 small ViT models. The optimizee MLPs are sampled from this hyperparameter distribution: $l \in [1, 2]$ hidden layers of size in range $[20, 40]$ and sigmoid or relu activation function. The optimizee ViTs have $l \in [1, 3]$ layers, $h \in [1, 3]$ heads, with hidden dimension in range $[16, 64]$, mlp dimension in range $[16, 64]$ and head dimension in range $[8, 16]$. The optimizee task is to train the model for 100 steps on batches of 64 image-class examples.

**Hybrid loss function to improve generalization:** To promote generalization, we use the random-scaling trick proposed by [47]. Mnemosyne's optimizer is meta-trained by gradient descent using Adam optimizer with learning rate $\eta = 3e^{-4}$ to minimize a combination of two loss functions. The first is the task loss given by the sum of optimizee losses in a truncated roll-out of 5 MNIST training steps. The other one is an imitation loss given by the mean squared error between Mnemosyne's updates and expert-optimizer (Adam) updates for same inputs. Importantly, this imitation loss is different from the one proposed in [13] which uses off-policy expert roll-outs for imitation. In our case, we provide expert supervision for the on-policy updates. This mitigates the problem of divergence from expert's trajectory, often observed in behaviour cloning. Our imitation loss acts as a regularizer which prevents Mnemosyne's optimizer from over-fitting on the optimizee task that it is trained on. We emphasize that expert's learning rate $\eta_{\mathrm{exp}} = 3e^{-2}$ **was not obtained via any tuning process**.

Our optimizer model has minimal input feature engineering and our meta-training setup is significantly simpler than those considered in the literature [49, 13, 47, 71]. Even so, we can successfully apply Mnemosyne's optimizer to a variety of tasks due to its efficient memory mechanism. Furthermore, Mnemosyne's memory cells can be easily combined with any of the existing L2L methods that use LSTMs for memory-encoding.

**Results:** After meta-training, Mnemosyne's optimizer was tested on NN training tasks with different NN architectures and datasets. Recall that Mnemosyne only saw one ML task of MNIST classifier training for 100 steps during meta-training. Fig. 10 shows that Mnemosyne can optimize MLPs with different NN archtitectures and activation functions on MNIST image classifier training. Note that, Mnemosyne converges significantly faster than popular analytical optimizers, RMSprop and Adam while retaining similar asymptotic performance. Mnemosyne can train NNs for long horizons of thousands of steps while baseline LSTM optimizer [2] struggles to minimize classification loss beyond a few hundred steps.

**Transformers:** The results were already presented in the main body of the paper (see: Sec. 5.1). We want to add that, as for experiments from Fig. 10, here Mnemosyne's optimizer is faster than standard analytical optimizers and much more stable than LSTM optimizer. Fig. 11 shows the benefit of using expert imitation-loss for long-horizon stability of the Mnemosyne's optimizer.

Our results on training Transformers with Mnemosyne naturally lead to the question of the role that Transformer-based optimizers can play in training Transformers architectures. It is well known that Transformer training requires nontrivial optimization techniques [45], e.g. learning rate schedulers (for that reason SGD was replaced with Adam in Transformer-training). Furthermore, for larger architectures training is slow, often prohibitively (unless the model is trimmed down, for instance by replacing long-range attention modeling with local attention of the controllable attention radius). Attention-based optimizers can potentially address this problem, since they improve convergence (and thus effectively reduce training time) even if meta-trained on much simpler tasks as we show in Fig. 3.

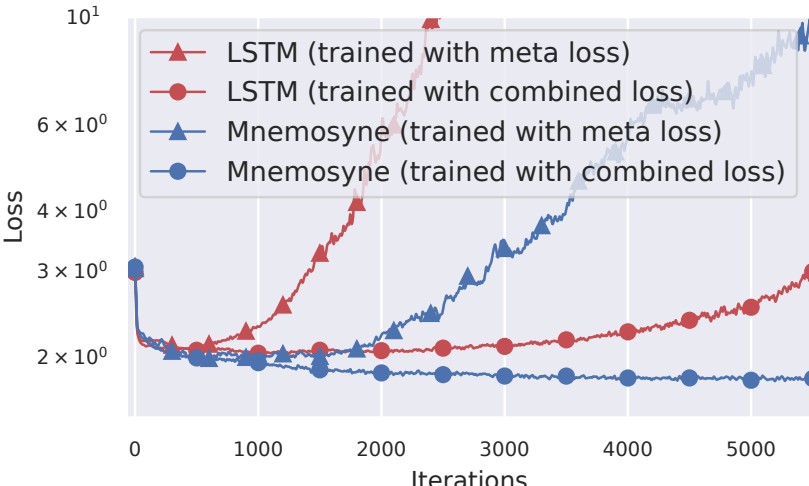

Figure 11: Impact of training the optimizer with combined meta loss and imitation loss can be seen in generalization to a long horizon rollout. All variants were trained only on length 100 rollouts.

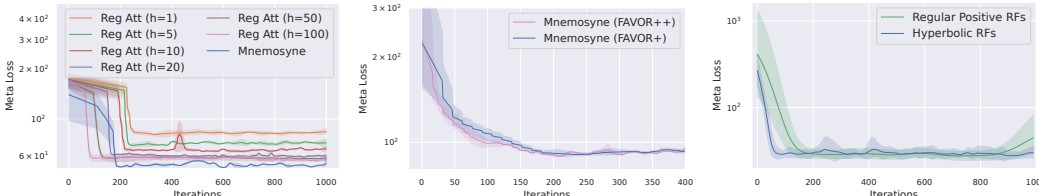

Figure 12: Ablation Studies. **Left:** Comparison of the Mnemosyne's linear CAM with regular attention memory blocks with different history cache lengths ($h$). **Middle:** Meta-training curves of Mnemosyne optimizer with FAVOR+ and FAVOR++ mechanism for CAM. **Right:** Meta-training curves of Mnemosyne optimizer with different kernel transformation functions for CAM.

## B.2 Mnemosyne's CAM mechanism vs regular attention

We have tried to use regular Transformer blocks to encode associative memory for Mnemosyne's temporal module. For applying regular attention to online optimizer autoregressively, a limited-length cache of historical gradients has to be maintained. A self-attention map over the history sequence is generated and used to encode memory. Fig. 12 (left) shows the meta-training curves for regular attention optimizers with different history cache lengths. As we increase the cache length, the performance improves and the memory requirement scales quadratically. Due to this limitation, we could not implement a regular attention based optimizer with cache length more than 100. On the other hand, Performer's memory cell defining CAM can attend to theoretically unbounded history and out-performs regular attention variants with fixed memory requirement.

## B.3 Different RF-mechanisms: detailed look

Fig. 12 (middle) compares the performance of Mnemosyne's optimizer applying FAVOR+ and FAVOR++ mechanisms in CAM. FAVOR++ mechanism provides strongest theoretical guarantees for the capacity of the associative memory model. It also leads initially to faster convergence, but asymptotically performs similarly as the FAVOR+ variant. Due to the simpler implementation of FAVOR+, we use it for all experiments with Mnemosyne's optimizer.

Optimizers with both regular positive and hyperbolic random features kernel learn similarly, but the latter has much lower variance (see: Fig. 12 (right)) and thus it became our default choice.

## B.4  Ablations over different kernel functions in CAM mechanism

In Fig. 13, we compare different kernel functions for linear attention in CAM. Our ablation study shows that FAVOR+ method outperforms other functions. This informs the choice of the kernel in all our experiments.

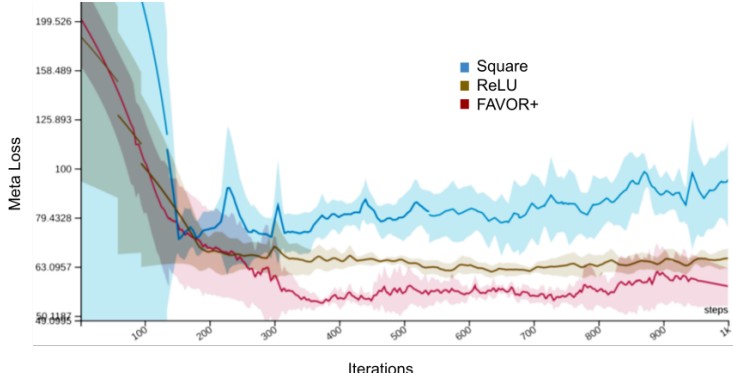

Figure 13: Comparison of the coordinate-wise Mnemosyne with ReLU kernel, square kernel and FAVOR+ approximation of the softmax kernel. We present the meta-training loss of the three variants. FAVOR+ mechanism outperforms other variants.

## B.5  Ablations over different discount factors and number of RFs in CAM mechanism

In Fig. 14, we present detailed ablation studies over discount factors $\tau$ as well as the number of random features applied by our default CAM mechanism leveraging hyperbolic cosine random features.

## B.6  Benchmarking different depths of the temporal module

Finally, we run ablations over different number of temporal encoders in Mnemosyne's temporal block. We noticed that modest increase of the number of encoders improves loss in meta-training and meta-training very deep variants is particularly challenging (as requiring much more data). Since in this paper we decided to use simple meta-training strategies and furthermore increasing the number of temporal encoders did not lead to substantial gains, we decided to choose shallow temporal encoders' architectures. The results are presented in Fig. 15.

## B.7  Compute Resources Used

All Mnemosyne optimizer variants were trained and tested on a TPU pod containing $4$ TPU v3 chips with JAX. Hundreds of rounds of training and inference were needed to compare different variations, tasks and meta-losses.

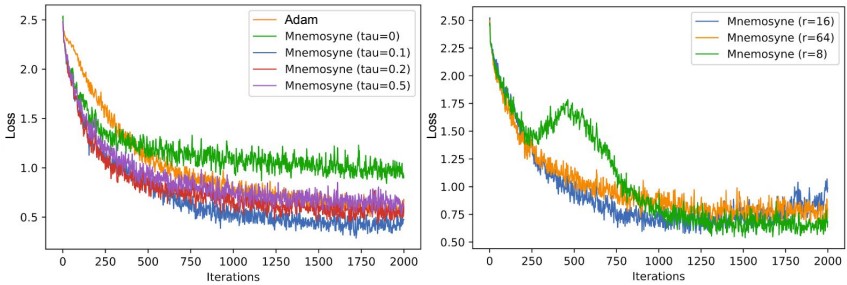

Figure 14: **Left:** The comparison of Mnemosyne applying different discount factors with $\mathrm{Adam}$ optimizer in meta-training (MLP optimization). **Right**: The comparison of Mnemosyne applying different number of random features in the hyperbolic cosine random feature mechanism used in CAM.

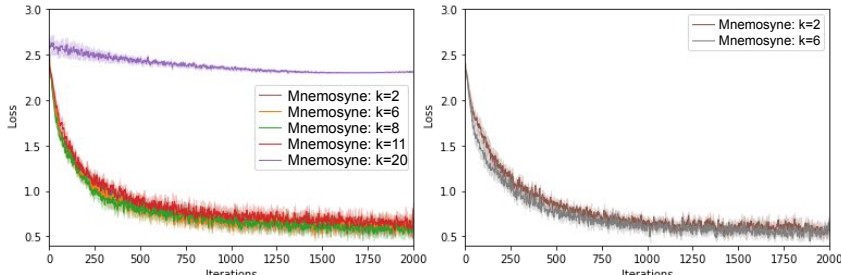

Figure 15: Comparison of the meta-training loss for Mnemosyne variants applying different number of temporal encoders $k$. Since several variants on the left figure performs similarly, on the right figure we highlight top two. The meta-training is conducted on the MLP optimization tasks and MNIST data.

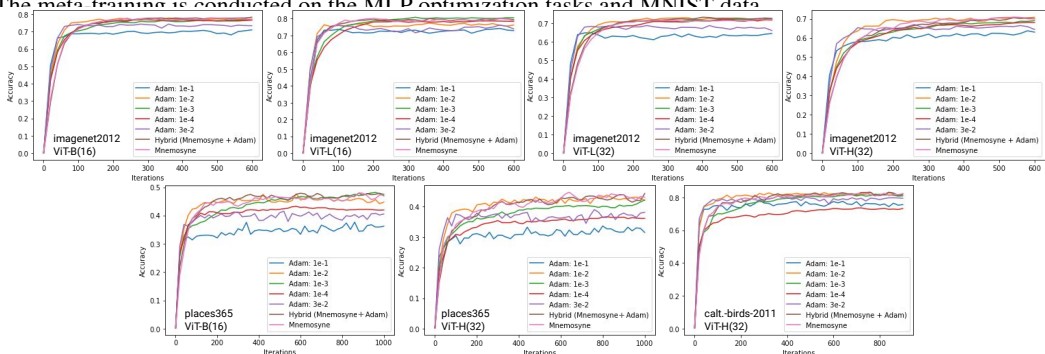

Figure 16: Coordinate-wise Mnemosyne across different ViT architectures and datasets, as described in Sec. B.8.1. Mnemosyne matches or outperforms optimal $\mathrm{Adam}$ variants without any hyperparameter tuning.

## B.8 Coordinate-wise Mnemosyne versus hard-coded optimizers for larger ViTs

### B.8.1 ViT last-layer fine-tuning

In this study, we benchmarked Mnemosyne on different sizes of ViT architectures: ViT-Base, ViT-Large and ViT-Huge (ViT-B(x), ViT-L(x) and ViT-H(x) respectively, where $x$ defines the patch size), see: Tab: 2. We used the coordinate-wise variant of the Mnemosyne. We run tests on the following datases: $\mathrm{imagenet2012}$, $\mathrm{places365}$ and $\mathrm{caltech}\text{-}\mathrm{birds}\text{-}2011$. We were optimizing the last layer of the ViT-architecture and used $\mathrm{Adam}$ expert with learning rate $\eta = 3e^{-2}$ as a regularizer (see: our discussion above on meta-training). The learning rate was not tuned in any way. In fact (as we show below) $\mathrm{Adam}$ optimizer applying this learning rate is characterized by the sub-optimal performance. We tried two versions of Mnemosyne: (a) a variant that solely optimizes the last layer of ViT (reported in the main body) and (b) the *hybrid* variant, where Mnemosyne is used to optimize the weight-matrix of the last layer and $\mathrm{Adam}$ with learning rate $\eta = e^{-3}$, to optimize the bias vector. That learning rate was also not tuned in any particular way and, as before, if applied purely within $\mathrm{Adam}$, produces sub-optimal results. The purpose of that last experiment was to assess how efficient the strategy of optimizing jointly with Mnemosyne and a hand-designed optimizer is. The results are presented in Fig. 16 and Fig. 17. We see that: (a) Mnemosyne without any hyperparameter tuning matches or outperforms optimal $\mathrm{Adam}$ variants, (b) it also substantially outperforms $\mathrm{Adam}$ variant used as an expert in meta-training. This is valid for both: regular Mnemosyne as well as the hybrid version.

### B.8.2 ViT multi-layer fine-tuning

Here, we used a *light* version of coordinate-wise Mnemosyne using a single temporal encoder layer with hidden dimension $8$. This reduced the memory requirement of the Mnemosyne optimizer state. We fine-tune ViT-B model on CIFAR-100 dataset with batch size $128$. We were able to fine-tune last 2 transformer layers along with the embedding, cls and head layers with Mnemosyne. Rest of the model was fine-tuned with $\mathrm{Adam}$ (learning rate $= 1e^{-3}$). For comparison, the same baseline $\mathrm{Adam}$ variant is to fine-tune the complete model.

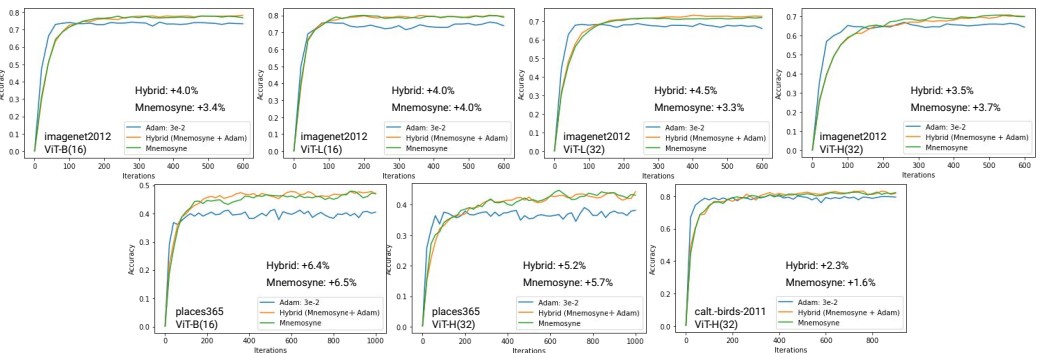

Figure 17: The results from Fig. 16, but narrowed down to the comparison between two Mnemosyne variants and Adam optimizer applying learning used in meta-training of these variants of Mnemosyne. The expert substantially underperforms in all the cases (we explicitly put the gains coming from the two variants of Mnemosyne as compared to the expert variant). This shows that Mnemosyne does not learn to imitate the expert.

Table 2: Hyperparameters for the different ViT models used in this paper

| Model | Heads | Layers | Hidden Dim. | MLP Dim. | Params | Patch Size |
|---|---|---|---|---|---|---|
| ViT-Base | 12 | 12 | 768 | 3072 | 86M | 16 |
| ViT-Large (16) | 24 | 16 | 1024 | 4096 | 307M | 16 |
| ViT-Large (32) | 24 | 16 | 1024 | 4096 | 307M | 32 |
| ViT-Huge | 32 | 16 | 1280 | 5120 | 632M | 32 |

## B.9 Tensor-wise Mnemosyne versus hard-coded optimizers for ViT-H

We finetuned the embedding and cls layer of ViT-H (see Tab: 2 for hyperparameter) using tensorwise ($\sim 1M$ params), while the head was trained using Adam. The rest of the transformer parameters are fixed to the pre-trained value for all methods. The batch size was set at 128 for all methods.

## B.10 Super-Mnemosyne: combining coordinate- and tensor-wise strategies for ViTs

We finetuned the top-8 layers of the ViT-Base model (see Tab: 2) along with the head, cls and embedding layer before we ran out of memory ie $\sim 50M$ parameters with a batch size of 256. Large tensor such as: a) the MLP block withing each layer, b) the head layer was finetuned using lite version of coordinate-wise. Rest of the tensors were finetuned using tensorwise. The bottom 4 layers of the model were kept fixed for Mnemosyne. For Adam baselines we finetuned all layers.

## B.11 BERT-pretraining NLP Transformers with Mnemosyne

We trained the Bert base model, whose Hyperparameters are shown in Tab: 3. The details of the training dataset used is shown in Tab: 4. We trained all parameters from scratch for all methods, with a batch size of 512. For the Mnemosyne results shown in Fig: 7, we trained all parameters except the token embedding using Tensorwise Mnemosyne ($\sim 86M$ parameters). The token embedding was trained using Adam with learning rate $1e - 4$. For Adam baseline we trained all parameters.

Table 3: Hyperparameters for the Bert base model

| Model | Heads | Layers | Hidden Dim. | MLP Dim. | Params | Compute | Loss |
|---|---|---|---|---|---|---|---|
| Bert-Base | 12 | 12 | 768 | 3072 | 110M | 4x2 TPUv3 | MLM |

## B.12 Soft prompt-tuning massive T5XXL Transformers with Mnemosyne

We use coordinate-wise Mnemosyne to prompt-tune [38] T5XXL model [55] (see Table 5 for hyper-parameters) on SuperGLUE benchmark. Batch size 32 was used. The length of the soft-prompt sequence was 30 and each soft-prompt vector was of size 4096, making the total number of trainable parameters 122880.

Table 4: Dataset used for pre training.

| Dataset | # tokens | Avg. doc len. |
|---------|----------|---------------|
| Books [77] | 1.0B | 37K |
| Wikipedia | 3.1B | 592 |

Table 5: Hyperparameters for the T5XXL model

| Model | Encoder Layers | Decoder Layers | Heads | Head Dim. | Embedding Dim. | MLP Dim. | Params | Compute |
|-------|----------------|----------------|-------|-----------|----------------|----------|--------|---------|
| T5XXL | 24 | 24 | 64 | 64 | 4096 | 10240 | 11B | 2x2x4 TPUv3 |

## C  Mnemosyne for training initial optimization conditions

Mnemosyne can be also applied to learn initial conditions for the optimization.

In this section, our considered model of using $g_\theta$ is more general than the one presented in Eq. 1 and is of the form given below for $g_\theta = (g_{\theta_1}^{\text{init}}, g_{\theta_2}^{\text{hist}})$, $\theta = [\theta_1; \theta_2]$ and a *context-vector* **c**:

$$\begin{cases} \mathbf{x}_0 = g_{\theta_1}^{\text{init}}(\mathbf{c}), \\ \mathbf{x}_{t+1} = g_{\theta_2}^{\text{hist}}(f, \mathbf{x}_0, ..., \mathbf{x}_t) & \text{if } t > 0 \end{cases} \tag{38}$$

The optimizer $g$ is now explicitly split into two-parts: (1) $g_{\theta_1}^{\text{init}}$ that learns initial optimization point from the context-vector, e.g. the image encoding the scene (as it is the case in learnable-MPC setting [72]), and (2) $g_{\theta_2}^{\text{hist}}$ that processes the history of the optimization steps, as we described before. Depending on the application, either one or the other optimizer (or both) are turned on. Critically, both are encoded as light scalable Transformers. Optimizer $g_{\theta_1}^{\text{init}}$ applies spatial bi-directional attention while $g_{\theta_2}^{\text{hist}}$ uses the spatio-temporal variant, described in the main body of the paper.

Below we show how this paradigm can be used in Robotics to learn initial points for the MPC optimization.

### C.1  Spatial Mnemosyne for initializing MPC optimizers

**Preliminaries:** We applied Mnemosyne with bidirectional spatial attention for learning trajectory optimizers to be used for wheeled robot navigation in complex, photo-realistic simulated environments. The robot uses vision sensors for observing an occupancy grid of the environment and navigates using linear and angular velocity control. Navigation in challenging environments requires efficient high-speed robot control. Model Predictive Control (MPC) presents an efficient approach to the navigation problem, provided that the motion-planning with the environment model can be carried out within computational real-time limits [72]. Motion planning with efficient trajectory optimizers such as iterative Linear Quadratic Regulator (iLQR) [41] is one way to implement MPC. However, in challenging layouts with narrow corridors and doors and with conservative safety and collision avoidance constraints, iLQR struggles to converge to the optimal trajectory. Sequential Quadratic Programming (SQP) [62] is often a more robust optimizer that can handle difficult non-linear constraints. However, SQP is significantly, sometimes $\sim 10$ times, slower than iLQR and hence cannot be deployed in real-time settings. Deep Learning models have been shown to accelerate SQP by warm starting the optimization process [30]. We learn Mnemosyne's optimizer to imitate SQP

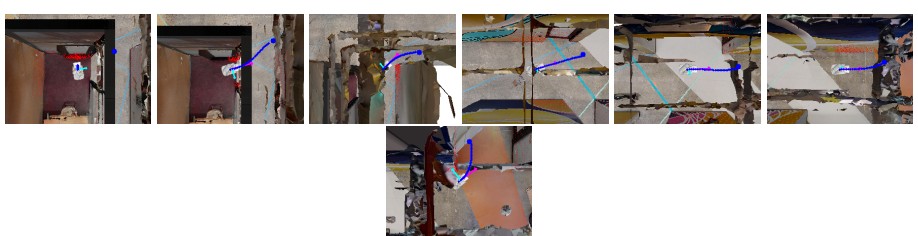

Figure 18: Navigation in a photo-realistic environment with MPC policy using Mnemosyne optimizer for initializing SQP solver.

behaviour and initialize it with an approximately optimal trajectory as a starting point from which SQP refines to an optimized and feasible trajectory.

**Training details:** Mnemosyne's optimizer, $g_{\theta_1}^{init}$, receives current robot pose $p_r$, a goal pose $p_g$ and a visual occupancy grid as the context, $\mathbf{c}$. The occupancy grid is processed by an image encoder module, a ViT where the attention mechanism is approximated by bidirectional Mnemosyne memory. As in a ViT, the occupancy grid is first pre-processed by a convolution layer and then flattened to a sequence. Each element (token) of the sequence corresponds to a different $5 \times 5$ patch of the original frame which is then enriched with positional encoding. The pre-processed input is then fed to $3$ Mnemosyne attention and MLP layers of hidden dimension $64$. The final embedding of one of the tokens is chosen as a latent representation of the occupancy grid, $l_{oc}$. $p_r$, $p_g$ and $l_{oc}$ are concatenated and processed by an MLP which outputs the predicted action trajectory, $\mathbf{x}_0$.

An offline dataset of SQP optimization examples is collected by running MPC navigation agent in $2787$ different environments. Each navigation run has $180$ steps on average. For each MPC step, one instance of trajectory optimization with SQP was run and a pair of input context $\mathbf{c}$ and the final optimal trajectory $\mathbf{x}_T$ was recorded. A total of $\sim 500,000$ training examples were collected. Mnemosyne's optimizer was trained with supervised learning on the SQP dataset by minimizing mean squared error between the SQP optimal trajectories and the predicted trajectories.

**Results:** After training, the predicted trajectory from Mnemosyne was used to initialize SQP optimization. Without Mnemosyne initialization, SQP optimization was capped at maximum $10$ iterations. It took on average $4.78$ iterations and $0.12$sec for the SQP solution to complete. With Mnemosyne initialization, SQP is only run for $1$ iteration to reach the optimal trajectory. SQP generates a trajectory that satisfies kinematic, dynamic and safety constraints for the robot which transformer alone e.g. [7] cannot natively guarantee. This reduces the optimization time by more than half to $0.048$sec on average which is under real-time constraint. It includes $0.011$sec for Mnemosyne inference and the rest for SQP iteration. A sequence of snapshots during navigation with Mnemosyne-SQP optimizer in a sample environment is shown in Fig. 18.

