# OpenReview forum: "Mnemosyne: Learning to Train Transformers with Transformers"
_NeurIPS.cc/2023/Conference — NeurIPS 2023 poster_

### Official Review · Reviewer_JvEY · 2023-07-07

**Soundness:** 3 good
**Presentation:** 3 good
**Contribution:** 3 good
**Rating:** 6
**Confidence:** 4

**Summary:**

This paper proposes to use transformers to learn optimization policies for training deep learning models. Here, the sequence corresponds to the state sequence (representing the internals of the model) in the trianing process. The intuition is that when we collect many sequences from many training processes, we can extract knowledge from these sequences via using transformers.


**Strengths:**

It appears to be a quite challenging project. Merely to make this kind of meta-learning works seem to be already very compute-heavy. Also, the vanilla version of the transformer has quadratic space and time complexity so it will be slow.

This work developed a quite large collections of careful hacks to make the idea work. For example, performance that uses low-rank matrix to approximate attention, and additional pooling is performed to further trim down computations. There are also other low level tricks like using exponential moving average.

There are also some theoretical results related to associate memory.

The sheer amount of work seems sufficient for a neurips acceptance.


**Weaknesses:**

I honestly cannot quite interpret the associate memory part. It seems to suggest the proposed architecture has sufficient expressive power. I thought the more important question is to understand why

Also, the lack of source code seems a minus. In general, I feel the quality of source code plays an increasingly important role in final decision, e.g., you can clearly see some system effort is more serious than others.


**Questions:**


- When we do this kind of meta learning, does that mean that the model states need to be stored for every batch? That seems very expensive.
- Can you comment environment we need to run the proposed solution, e.g., a few GPUs & a few days or 1000 A100s for a month?

---

> ### Author Rebuttal · Authors · 2023-08-08
>
> Thank you very much for the review and all the comments.We address them in detail below. We are happy to hear that the Reviewer appreciates the amount of work conducted in this work and evaluates it as sufficient for NeurIPS acceptance.
>
> > *cannot quite interpret the associate memory part*
>
> Our theoretical results show that the applied memory mechanism is capable of storing (on average) optimization history of the exponential length (as it acts as the large capacity associative memory model).
> We have run additional tests comparing the attention linearization mechanism with the popular Performer-ReLU and Performer-square variants. We have not put them in the original manuscript due to space constraints, but refer to them in the rebuttal (see the pdf attached to our global comment). Our results show that sub-exponential (on average) storage capacity of the associative memory model related to the ReLU and square variants does not admit the theoretical analysis from Sec. A.2 hurts performance. As summarized above, our theoretical results from Sec. A.2 show that the provided memory mechanism is capable of storing (on average) exponential number of points from the optimization history, effectively enabling the model to take into account much longer optimization history and consequently, improving overall optimizer’s performance.
>
> We want to emphasize that our theoretical analysis critically relies on the fact that considered random feature map mechanisms approximate the softmax kernel. Therefore it clearly distinguishes those methods from other random feature map mechanisms (also applied in Performers) that linearize non-softmax kernels. As we explain above, this distinction is confirmed by the additional conducted experiments that we include in the rebuttal. Thus the importance of our theoretical analysis lies in the fact that it leads to the choice of the right linearized associative memory mechanism without any additional ablation studies.
>
> In the final version of the paper, we will make those theoretical insights much more explicit in the main body of the paper and include aforementioned results on comparing ReLU- and square- variants with the hyperbolic cosine one.
>
>
> In the rebuttal, we have added the comparison of our associative memory model with the optimizers replacing regular LSTM memory unit with the powerful S4 memory unit based on [state space machines](https://arxiv.org/pdf/2111.00396.pdf), as an alternative to LSTMs (and Transformers). Our results are presented below:
>
> Experiment 1: imagenet-ViT-H(patch size = 14), last layer fine-tuning (500 iterations):
>
> | Methods | Accuracy(%) |
> |--------------|:-----------:|
> | Mnemosyne | **72.3** |
> | S4 | 7.5 |
>
> Experiment 2: imagenet-ViT-H(patch size = 8), last layer fine-tuning (500 iterations):
>
> | Methods | Accuracy(%) |
> |--------------|:-----------:|
> | Mnemosyne | **81.4** |
> | S4 | 35.0 |
>
> Our experiments confirm our previous findings. We show in particular that S4-based architectures fail to fine-tune vision Transformer models under consideration, as opposed to Mnemosyne. Thus RNN-based approaches seem to underperform as compared to attention-based methods for the learnable optimization.
>
> > *the lack of source code*
>
> Thank you for the comment. We plan to publish the code upon the acceptance of the paper. We did our best to make sure that *all the details* are included in the paper and put many of them in the Appendix, in particular:
>
> 1.  detailed discussion of the meta-training phase: we believe we have provided all the details regarding meta-training in the meta-training paragraph of Sec. B.1: (a) the process for creating mlp and vit tasks for meta-training (l. 760-764), (b) number of iterations used in meta-training (l. 764), (c) the particular form of the hybrid-loss function, including Adam regularizer with the explicitly given non-tuned learning rate (l. 765-776);
> 2.  ablations studies over varying tau-parameters for the exponential discount strategy (Fig. 13 - left), leading to our default choice of tau;
> 3.  ablation studies over different number of applied random features r, leading to our default choice of r (Fig. 13 - right);
> 4.  ablation studies over Mnemosyne architectures with varying number of temporal encoders (Fig. 14), leading to the conclusion that shallow encoder modules suffice for Mnemosyne;
> 5.  used computational resources (Sec. B.6);
> 6.  hyperparameters for different models that Mnemosyne was tested on in the paper (Table 1);
> 7.  hyperparameters for the base BERT models that Mnemosyne was tested on in the paper (Table 2);
> 8.  hyperparameters for the T5XXL models that Mnemosyne was tested on in the paper (Table 4).
>
> > *When we do this kind of meta learning, does that mean that the model states need to be stored for every batch? That seems very expensive. Can you comment on the environment we need to run the proposed solution, e.g., a few GPUs & a few days or 1000 A100s for a month?*
>
> Thank you for the comment. All Mnemosyne variants were trained and tested on a TPU pod containing 4 TPU v3 chips with JAX. Mnemosyne is meta-trained on an MNIST classification task with 3 small MLP and 3 small ViT models. This makes the meta-training very efficient. Saving optimizer states for these small models has a very small memory and computational footprint. A meta-training run takes less than an hour on average. More details are given in the Appendix.

---

### Official Review · Reviewer_xFuP · 2023-07-14

**Soundness:** 3 good
**Presentation:** 3 good
**Contribution:** 3 good
**Rating:** 6
**Confidence:** 2

**Summary:**

The paper introduces Mnemosyne, a new learnable optimizer that can train neural networks without task-specific tuning. The authors claim that it employs minimal resources and handles larger parameter sets, outperforming LSTM optimizers and matching state-of-the-art ones. Its efficacy is demonstrated in the fine-tuning of Vision Transformers, BERT models, and large T5XXL models. In addition, it also provides the theoretical analysis of the compact associative memory used by the proposed Mnemosyne.

**Strengths:**

（1）The proposed Mnemosyne introduces a novel approach by utilizing spatio-temporal low-rank implicit attention Transformers to create learnable optimizers. This innovative idea goes beyond traditional optimizers by eliminating the need for task-specific tuning in training neural networks. The idea seems interesting.

（2）Mnemosyne has demonstrated superior performance in various use cases, such as fine-tuning Vision Transformers, pre-training BERT models, and soft prompt-tuning large T5XXL models. This shows its broad applicability and robustness

**Weaknesses:**

(1) The method's effectiveness on broader model training remains unknown as it only trains or fine-tunes a subset of Transformer layers. Moreover, given the Transformer's huge data requirements, its performance on larger datasets is also unclear.

(2) As the method only trains or fine-tunes some layers of certain Transformer architectures, its title's emphasis on training Transformers seems somewhat overstated.

**Questions:**

See the weakness.

**Limitations:**

The authors have discussed the limitations.

---

> ### Author Rebuttal · Authors · 2023-08-08
>
> Thank you very much for the review and all the comments. We address them in detail below.
>
> > *The method's effectiveness on broader model training remains unknown as it only trains or fine-tunes a subset of Transformer layers*
>
> We would like to clarify that our experiments involve in particular pre-training *full base BERT model with 86M parameters* rather than thee subset of model’s parameters (l.333-345), which to the best of our knowledge was never done before with learnable optimizers (due to space complexity challenges) as well as several (rather than one) layers of ViT models (l.345-357). The results in Sec. 5.4 are for training *top 8 (out of 12) layers* (*~50M parameters*) of the ViT-Base model with the Super-Mnemosyne variant. Those examples best illustrate the effectiveness of the spatial encode (and consequently, tensor-wise Mnemosyne)r which is one of the critical contributions of this work and can be used to overcome memory issues of the learnable optimizers relying on the coordinate-wise strategy. To summarize, we do think that showing that learnable optimizers can be successfully applied on such a scale is a significant contribution that was not achieved before.
>
> We do not claim that our approach using spatial encoding resolves all memory issues of learnable optimizers. But we do think that it is an important step towards scaling up learnable optimizers to massive Transformer-models, as allowing to train architectures of sizes that were beyond the reach of the proposed before learnable optimizers.
>
> Furthermore, fine-tuning a subset of layers or soft prompt-tuning massive models (in the paper we successfully soft prompt-tune 11B+ T5XXL models, see: l.323-330) are very popular techniques of customizing large models. This is the case since pre-training from scratch or fine-tuning all the parameters is not possible without extensive computational resources. That is why we do think that our work is directly relevant to researchers leveraging modest computational resources which is well within the scope of the ML democratization efforts. We do not think this research is as important as training larger and larger models. The research on training a subset or a more compact set of parameters is actually one of the most rapidly growing areas of the present efforts on Transformers and led to several important methods proposed recently (such as [LORA](https://arxiv.org/abs/2106.09685) or [adapters](https://arxiv.org/abs/1902.00751)). Those methods have been proven to be competitive with full model fine-tuning for many downstream tasks, yet they clearly require much smaller computational resources. Even more recently (after our paper was written), the mechanism of [SubTuning](https://arxiv.org/pdf/2302.06354.pdf) was introduced. The authors explicitly state (in the context of fine-tuning) that :
>
> *"...We observe that not all layers are created equal: different layers across the network contribute variably to the overall performance, and the optimal choice of layers is contingent upon the downstream task and the underlying data distribution…”*
>
> We do hope that our response addresses all the concerns raised by the Reviewer regarding the broader impact of our method.
>
> >*…given the Transformer's huge data requirements, its performance on larger datasets is also unclear*
>
> Our temporal modules consists of at most two temporal encoders (in most applications just one), since our ablation studies showed that deeper architectures provider negligible gains (see: Fig. 14) given small set of training tasks (which we wanted to keep to make the entire meta-training procedure simple). Thus the underlying Mnemosyne’s encoding entire memory mechanism is compact. We ended up applying very simple meta-training on a couple of toy-size MLP and ViT tasks (see: meta-training paragraph in Sec. B.1). This was the case also when the target downstream task was not related to the meta-training tasks, e.g. for base BERT pre-training or T5XXL soft prompt-tuning. In those cases Mnemosyne was operating on the 86M and 11B+ size models respectively. We did not observe any need to scale up meta-training to a larger set of tasks to efficiently optimize larger models. That was also the case when we increased the number of fine-tuned ViT layers (see: l.345-357 in Sec. 5). The only modification of meta-training was introducing larger weight-matrices in meta-training (so that the spatial encoder learns how to encode them), yet the sheer number of tasks and their difficulty remained the same.The entire meta-training consisted of only 100 iterations. All these observations clearly show that Mnemosyne is not data-hungry in meta-training and meta-training does not become a bottleneck when downstream optimization tasks become bigger. We will clarify it in the final version of the paper, by providing a separate paragraph on the scalability of meta-training in the main part of the paper.

---

> > ### Comment · Reviewer_xFuP · 2023-08-16
> > **Response to the authors**
> >
> > I am not an expert in the relevant field, but after reading the author's response and considering the opinions of other reviewers, I decided to keep the original score.

---

### Official Review · Reviewer_V1vR · 2023-07-31

**Soundness:** 3 good
**Presentation:** 2 fair
**Contribution:** 2 fair
**Rating:** 5
**Confidence:** 4

**Summary:**

The paper proposes to use transformers to learn to optimise. The proposed architecture relies on the performer architecture and leverage attention to encode long-term dependencies. The authors discuss two variants, a coordinate-wise and tensor-wise approach, to account for the dependencies between the parameters to optimise. They also propose to use a hierarchal pooling strategy to bypass the linear attention of performers. The authors compare their method to classical optimizers, such as ADAM, and LSTM-based optimisers. The paper also provides a theoretical analysis of the compact associative memory used by the method.

**Strengths:**

The authors propose a transformer-based optimiser by combining a topological and a temporal encorder. The resulting transformer is outperforming LSTM-based counterparts on the experiments conducted by they authors. The method is also competitive with widely used optimisers such as ADAM. The overall results are encouraging and the reported results cover both NLP and CV applications.

**Weaknesses:**

While the authors propose a transformer-based architecture to learn an optimiser, their choices are mainly driven by practical considerations. The authors do not motivate their choices beyond these and do not discuss trade-offs and/or other possible choices. The methodological contributions of this work are modest beyond combining existing components and the use of a hierarchical pooling strategy. While I have no doubt that making these components work well together is non trivial, the challenges to make them work together are not discussed. Hence, it appears that the paper does not do justice to work that has been done in the background. More importantly, the lack of details would make it difficult for other researchers to reproduce the results.

The authors provide a theoretical analysis of the compact associative memory used, but the importance and implications of this result is not discussed.

While the experiments cover the NLP and CV domain, reported results were underwhelming. The authors conducted first small scale experiments where they compared to a wide range of off-the shelve optimizers. However, in the NLP and CV domain, they only optimised a subset of the parameters with the proposed approach, cast doubts on the broad applicability, its scaling properties and its robustness.

**Questions:**

I would like to ask the authors the following questions:

1/ Can you provide a summary of the meta-training procedure and a discussion of the choices made? How robust is Mnemosyne to the specific procedure used and the set-up?

2/ Can you explain why long-term dependencies are important when learning an optimizers? Can you provide evidence that the proposed architecture is able to capture these and leverage them?

3/ Can the authors articulate the importance of the theoretical analysis?

4/ Why did you consider the VeLo architecture in the experiments? Did you consider other options?

5/ Can you provide details of the configuration of Mnemosyne in the experiments? How many of these did you try out and how did the results vary?

**Limitations:**

I would have liked the authors to be more vocally self-critical about the pros and cons of their proposed method. The paper reads like the approach is a sliver bullet, which obviously is not the case. I would have liked to see a discussion of the strengths and weaknesses of the method, an analysis of the robustness, an empirical validation of the claims (like importance of long-term dependencies), etc.

---

> ### Author Rebuttal · Authors · 2023-08-08
>
> >*choices driven mainly by practical considerations…implications of the theoretical analysis*
>
> See "Implications of theoretical analysis on design choices" in the global rebuttal.
>
> > *discussion of other choices*
>
> We chose Adam for non-learnable and LSTM for learnable optimizer baselines due to their popularity. See "New experiments: Comparison with other learnable optimizers" in the global rebuttal for experiments with more baselines.
>
> > *discussion of tradeoffs, self-criticality*
>
> See "Discussion of limitations" in the global rebuttal.
>
> > *modest methodological contribution…combining existing components…the challenges to make them work together are not discussed…*
>
> We justify that our contributions go beyond *“combining existing components”* as follows:
> 1. Implementation of linear attention in auto-regressive (rather than bi-directional) setting has not been done before. It requires algorithmic modifications like exponential discount strategy (l. 169). For auto-regressive FAVOR++, additional algorithmic ideas were needed (see: l.679-692 and "Implications of theoretical analysis on design choices" in the global comment).
> 2. Our hierarchical pooling encoder (HPE) mechanism in the tensor-wise Mnemosyne is *new*. HPE is different than other pooling techniques in Transformers (e.g. [Funnel-Transformers](https://arxiv.org/abs/2006.03236)), and relevant beyond LOs.
>
> Combining all these components was non-trivial (as the Reviewer noted) and we will address those challenges properly in the final version of the paper.
>
> > *the lack of details…a summary of the meta-training procedure*
>
> We plan to publish the code upon the acceptance of the paper. We did our best to include *all the details* in the paper and Appendix. (See "Summary of details in the Appendix" in the global comment). Mnemosyne is robust since our meta-training is simple by design. More extensive meta-training can improve the generalization of Mnemosyne but that study is out of scope for our paper.
>
> 1. Mnemosyne has been meta-trained on simple tasks (training toy-size MLP+ViT models), often not related to the downstream problem (full pre-training of base BERT 86M-parameter model or soft prompt-tuning of 11B+ T5XXL model etc.).
> 2. Different meta-loss functions were tried (with/without regularizers).
> 3. For the loss functions with regularizers, the hyperparameters of the regularization terms *are not tuned* at all.
>
> We kindly ask the Reviewer if they would like to see any other details. We are happy to provide them during the discussion phase.
>
> > *reported results were underwhelming…they only optimised a subset of the parameters*
>
> It is not true that we only optimized a fraction of parameters in all experiments. Our experiments include pre-training **full base BERT model with 86M parameters** (l.333-345), which was not done before with LOs (due to space complexity challenges) as well as several (rather than one) layers of ViT models (l.345-357). The results in Sec. 5.4 are for training **top 8 (out of 12) layers (~50M parameters)** of the ViT-Base model with the Super-Mnemosyne variant. We do think that showing that LOs can be successfully applied on such a scale is a significant contribution that was not achieved before.
>
> Furthermore, fine-tuning a subset of layers or soft prompt-tuning massive models (in the paper we successfully soft prompt-tune 11B+ T5XXL models, see: l.323-330) are very popular techniques of customizing large models. This is the case since pre-training from scratch or fine-tuning all the parameters is not possible without extensive computational resources. That is why we do think that our work is directly relevant to researchers leveraging modest computational resources which is well within the scope of the ML democratization efforts. The research on training a subset or a more compact set of parameters is actually one of the most rapidly growing areas of the present efforts on Transformers (as often providing competitive results) and led to several important methods proposed recently (such as [LORA](https://arxiv.org/abs/2106.09685) or [adapters]( https://arxiv.org/abs/1902.00751)). Even more recently (after our paper was written), the mechanism of [SubTuning](https://arxiv.org/pdf/2302.06354.pdf) was introduced. The authors explicitly state (in the context of fine-tuning) that :
>
> *"... not all layers are created equal: different layers across the network contribute variably to the overall performance, and the optimal choice of layers is contingent upon the downstream task and the underlying data distribution…"*
>
> Finally, our methods contribute to the field of stable long-range optimization with LOs (see: our response to the comment *"long-term dependencies"* below).
>
> > *long-term dependencies*
>
> See "Mnemosyne can perform stable long-horizon optimization" in the global rebuttal.
>
> > *VeLO*
>
> We chose to consider the VeLO architecture in our experiments for several compelling reasons:
> 1. VeLO is the SOTA in the LO literature, making it a pertinent benchmark.
> 2. VeLO’s feature engineering tricks and hierarchical model further strengthens it as a benchmark. These tricks improve the performance of the vanilla LSTM-based Los.
>
> Our objective was to show that we could improve VeLO’s performance by substituting the LSTM module with our compact associative memory (CAM) module. Note that for our experiments, we used a small-scale meta-training approach. This enabled us to clearly showcase one of the crucial differences between the LSTM and Mnemosyne approach to optimization: the former class of methods often struggle to generalize effectively to long-horizon evaluation tasks, while Mnemosyne mitigates this issue (see also: our discussion regarding long-term dependencies). By demonstrating the improvements achieved by simply replacing one memory mechanism with the other one, we highlighted the potential of the Mnemosyne’s memory model as an efficient mechanism for optimizing for long optimization horizons.

---

> > ### Comment · Reviewer_V1vR · 2023-08-21
> > **re: Rebuttal by Authors**
> >
> > I thank the authors for their detailed response. A couple of follow-up comments.
> >
> > 1/ I do not dispute the contributions mentioned above. However, these contributions seem to have been mainly driven by practical (to make sure the method could scale to at least modest size architectures), rather than methodological insights. if this is not the case I would love to hear the authors elaborate on these insights.
> >
> > 2/ It is great to hear the authors plan to make the code available. Arguably, I did not check the more detailed description of the implementation in the Appendix. I would encourage the authors to refer to the relevant Appendix in the main body of the paper. Regarding the robustness of the meta-training: (i) can the authors elaborate on what they mean by "Mnemosyne is robust since our meta-training is simple by design" and (ii) how can they claim this procedure is robust if they did not investigate the impact of the hyperparameters on it?
> >
> > 3/ I had missed the fact that authors had used entire Bert-base in the NLP experiments. My apologies. This being said, my point was that the method does not seem to be applicable to more recent neural nets (even though the authors put efforts and introduced novelties to scale it better). This suggest there are inherent practical limitations to the proposed method.
> >
> > Also, I do not dispute the fact that recent work focusses on parameter-efficient fine-tuning. However, i believe the proposed method is orthogonal to these works.

---

> > > ### Comment · Reviewer_V1vR · 2023-08-21
> > > **re: Rebuttal by Authors (continued)**
> > >
> > > (continuation from previous post; comments here follow the general response to authors.)
> > >
> > > 4/ Re Implications of theoretical analysis on design choices: Thanks for the additional clarifications. i would encourage the authors to ensure these details are included in the main body of the paper. What remains unclear is how the theoretical analysis motivates FAVOR++. Can you please elaborate?
> > >
> > > 5/ Re Mnemosyne can perform stable long-horizon optimisation: While appreciate the additional discussion provided by the authors, it does not answer my original questions which were (i) can you explain why long-term dependencies could be important when learning an optimiser and (ii) can you provide direct evidence that Mnemosyne is able to capture? Fig 10 and 11 show strong optimisation performance (which is great!), but it does not tell me that this strong performance is due to an ability to capture long-term dependencies.

---

> > > ### Author Response · Authors · 2023-08-21
> > >
> > > Thank you very much for the additional comments. We address them in detail below.
> > >
> > > 1. The whole field of learnable optimizers is definitely practically-motivated, since it aims to address the practical limitations of the brute-force counterparts. In that sense, our work is grounded in the practical considerations (e.g. scalability of the LOs; critical for the field). We do not consider this to be a weakness of the paper. However, as explained in detail in the rebuttal, we have developed several theoretical techniques that are interesting on their own, in particular:
> > >     * the autoregressive linear attention mechanism with smoothly decaying attention weights,
> > >     * scalable spatio-temporal Transformer model with a novel pooling algorithm,
> > >     * the autoregressive variant of the FAVOR++ mechanism (that prior to that paper was given only in the bi-directional form).
> > >
> > >     To the best of our knowledge, this is one of the few papers in the field with a rich set of techniques and the rigorous corresponding theoretical analysis.
> > >     We would like to kindly ask the Reviewer what methodological insights, orthogonal to any practical considerations the Reviewer would like to see in the paper.
> > >
> > > 2. Thank you very much for the comment !
> > >
> > >     (i) We would like to point out that one of the strategies in the L2L literature for improving performance is to scale-up meta-training to a large model close to the one seen in evaluation. This increases the tendency of LO to specialize on one task rather than be robustly applicable. On the contrary, we show that it is possible to restrict the meta-training budget and obtain LOs that can generalize to the training of models far larger than those seen in the meta-training. This highlights that the gradient update rules learned by our LO are generally useful rather than memorizing the loss landscape of the model to be evaluated on.
> > >
> > >     (ii) Unlike standard optimizers, Mnemosyne LO does not require any hyper-parameters (such as learning rate etc.) by design. In the paper, we have shown ablation studies for the design choices in the Mnemosyne architecture such as: (1) the decay factor (Fig 13. left subfigure), (2) number of random features (Fig 13. right subfigure), (3) number of temporal encoders (Fig.14), (4) different linear attention mechanisms approximating softmax attention (Fig. 12 middle and right subfigures) and (5) different linear attention kernels beyond softmax (in the rebuttal pdf). These choices are made before meta-training and based on the comparison studies. After meta-training, the Mnemosyne LO is applied without any downstream hyper-parameter tuning on the evaluation tasks.
> > >
> > > 3. Thank you very much for your comment. We do acknowledge the limitations of the method in terms of scaling up to fine-tuning all the parameters of the 1B+ size models. However we would like to emphasize that this was never a target of this manuscript and we leave it to future work. The scalability to massive models is a challenge for all LOs. We do believe that this work is a significant step towards addressing it, as capable of fine-tuning much larger sets of parameters than previously proposed techniques.
> > >
> > >     Given the above, parameter-efficient fine-tuning scenario is one of the most natural applications of the LOs and that is why we devote a considerable amount of time to discuss it in the paper.
> > >
> > >     We want to emphasize that the current applicability of the method is not related to how “recent” the neural network is, but to: how many parameters should be directly fine-tuned. All the architectures we applied our method to are currently used by the Community.

---

> > > > ### Author Response · Authors · 2023-08-21
> > > >
> > > > 4. Thank you for the comment. We will make sure to include all the relevant details in the final version of the paper. We have provided an explanation of the relevance of the presented theoretical analysis in the rebuttal and put it also here for Reviewer’s convenience.
> > > >
> > > >     Our theoretical results show that the mechanism is able to memorize long histories of the optimization trajectories, since its corresponding associative memory model has on average an exponential-size capacity (Theorem 4.3). Furthermore, our detailed choices for the particular linear attention variant approximating softmax-kernel are driven by the variance analysis of the corresponding estimators. In particular, our default hyperbolic cosine variant is characterized by the provably reduced variance and the autoregressive FAVOR++ mechanism provides additional accuracy gains. This theoretical analysis is perfectly aligned with the practical observations regarding the downstream performance of these methods (middle and right sub-figure of Fig. 12).
> > > >
> > > > 5. Thank you for a good question.
> > > >
> > > >     (i) In standard optimizer design, we observe the benefit from including time-series features that capture historical dependencies, such as first and second-order momentum in the case of ADAM. The LO field therefore focuses on memory based models such as RNNs for this task to enable the model to capture history features. It is well understood that attention mechanisms outperform RNNs in a wide range of sequence modeling tasks (the biggest example being language modeling). With this paper, we bring the same intuition to the field of LO.
> > > >
> > > >     (ii) Fig. 12 (left) shows the performance benefit of including longer gradient histories in the regular attention based LO input. Note that regular attention mechanism is memory and compute intensive and can only process small history sizes. This motivates the use of linear auto-regressive attention proposed by us. The plot shows that Mnemosyne’s CAM mechanism (which can theoretically attend to unbounded history) outperforms all regular attention LOs with fixed size history input. This is evidence for the importance of long-horizon historical context in the optimization task.

---

> > > > > ### Comment · Reviewer_V1vR · 2023-08-22
> > > > > **Re: Official Comment by Authors**
> > > > >
> > > > > I would like to thank the authors for their detailed comments and clarifications. I’m happy to increase my score.

---

> ### Author Response · Authors · 2023-08-18
> **A note to Reviewer V1vR**
>
> We would like to once more thank the Reviewer V1vR for the comments. We have addressed in detail all of them. Furthermore,  in our rebuttal we have added additional empirical results showing strong performance of our method. We would like to sincerely ask the Reviewer to read our response and consider changing the score.
>
> Thank you very much.
>
> The Authors

---

### Author Rebuttal · Authors · 2023-08-08

# New experiments: Comparison with other learnable optimizers
In the rebuttal we provide the comparison with two other classes of learnable optimizers:
1. Those using powerful [S4 memory units](https://arxiv.org/pdf/2111.00396.pdf) based on state space machines as an alternative to LSTMs (and Transformers).
2. The [Lion](https://arxiv.org/abs/2302.06675) family of LOs using a different approach of symbolic programming.

Our results are presented in the attached pdf. In the last experiment (Table 3), we skip S4 due to poor performance on the previous tasks. Mnemosyne is clearly the best among all three variants (in contrast to Lion, does not require hyperparameter tuning; also it applies simplistic meta-training: 4TPU v3 chips, see: Sec. B.1).

# Implications of theoretical analysis on design choices
Our paper provides much deeper theoretical insight than most others on learnable optimizers (LOs) (that focus on the empirical results). We have provided extensive theoretical and practical motivations for our design choices in the paper. We will make it more explicit in the final version:
1. The choice of the random feature (RF) mechanism is motivated by strong theoretical guarantees for the capacity of the corresponding associative memory (AM) (entire Sec. 4 and Sec. A.2). We are not aware of similar results for LSTM-based LOs. Other RF mechanisms that do not admit our theoretical analysis clearly underperform, as we show in the additional experiments in the rebuttal (see the attached pdf file). We compare there with Performer-ReLU and Performer-square RF mechanisms. Thus the importance of our theoretical analysis lies in the fact that it leads to the choice of the right linearized AM mechanism without any additional ablation studies.
2. Our default hyperbolic cosine RF mechanism (see: ll.186-191 and Sec. A.1) is motivated by the provably reduced estimation variance. We explicitly say it in l. 654-656.
3. We propose a novel autoregressive FAVOR++ (AFAVOR++) variant (see: Sec. A.1.1, l.679-692).
4. In Fig. 12 we show that: (a) hyperbolic cosine RFs are more robust than regular positive RFs (last plot), (b) AFAVOR++ provides faster convergence (middle plot).
5. We introduce novel exponential discount strategy (EDS) (l.169) to smoothly discount the past, theoretically motivated and compatible with Performers (see: Fig. 13 (left) for the detailed empirical verification of EDS).

# Summary of details in the Appendix
1. detailed discussion of the meta-training phase (meta-training paragraph of Sec. B.1), including:
	* the process for creating MLP and ViT tasks for meta-training (l. 760-764),
	* number of iterations used in meta-training (l. 764),
	* the particular form of the hybrid-loss function, including Adam regularizer with the explicitly given non-tuned learning rate (l. 765-776);
2. ablations studies over varying tau-parameters for the exponential discount strategy (Fig. 13 - left), leading to our default choice of tau;
3. ablation studies over different number of applied random features r, leading to our default choice of r (Fig. 13 - right);
4. ablation studies over Mnemosyne architectures with varying number of temporal encoders (Fig. 14), leading to the conclusion that shallow encoder modules suffice for Mnemosyne;
5. used computational resources (Sec. B.6);
6. hyperparameters for different ViT models that Mnemosyne was tested on in the paper (Table 1);
7. hyperparameters for the base BERT models that Mnemosyne was tested on in the paper (Table 2);
8. hyperparameters for the T5XXL models that Mnemosyne was tested on in thee paper (Table 4).

# Mnemosyne can perform stable long-horizon optimization
In our experiments, we made a noteworthy observation that Mnemosyne optimizers outperform their LSTM-based counterparts in terms of achieving a stable optimization phase over extended optimization horizons (see: Fig. 10 and Fig. 11). This results bears crucial importance since our goal is to achieve strong generalization from *small-horizon meta-training* to *long-horizon evaluation*, a notoriously difficult problem in the field of LOs. Prior attempts to address this challenge involved increasing the meta-training budget by including long-horizon tasks and made meta-training more complicated. Our approach takes a different direction that we do think about as novel. By incorporating a more expressive memory mechanism, we demonstrate that it is possible to learn highly generalizable optimization algorithms without the need of more complicated meta-training. We do believe that this finding opens up new possibilities for efficient and effective optimization in long-range scenarios and is an important advancement in the field. It also contributed to the efforts of democratizing ML.

# Discussion of limitations
1. Tensor-wise Mnemosyne is motivated by the limitations of coordinate-wise LOs (including coordinate-wise Mnemosyne). In Sec 3.2.1, we say:
	> *In that setting, even linear attention might not suffice. To address it, we introduce an additional hierarchical pooling mechanism, leading to the hierarchical pooling encoder (HPE).*
2. In the experiments (Sec. 5.3), we *explicitly* mention the limitations of the coordinate-wise Mnemosyne:
	>"Coordinate-wise Mnemosyne was not applicable here due to the memory footprint and thus it became a practical test for the memory efficient tensor-wise variant."
3. The tensor-wise variant is one way to address this problem, but it is not perfect. We say in Sec. 5.3:
	>"Although tensor-wise Mnemosyne can scale up to the full ViT model, here we freeze large tensors in the Transformer layers and fine-tune the rest. Training large tensors with tensor-wise Mnemosyne requires data and compute intensive large scale meta-training (to ensure that spatial encoders learn how to encode long sequences well). This will be the focus of the future work."
4. In the Broader impacts & limitations section we discuss how to further develop our meta-training strategies.

---

### Decision · Program_Chairs · 2023-09-21

**Decision:**

Accept (poster)

**Comment:**

This work proposes a transformer-based optimiser, which was shown to outperform LSTM-based counterparts. Experiments indicate that the learned optimizer is competitive with ADAM. Authors considered problems in both the NLP and the CV domain. The authors adequately addressed the concerns raised by the reviewers during the rebuttal and I would encourage the authors to incorporate the feedback they provided. The paper will be stronger by removing the ambiguities and imprecisions in the original manuscript.